# A decentralized approach for the aerial manipulator robust trajectory tracking

**Yarai Elizabeth Tlatelpa-Osorio**[1]*, **Hugo Rodríguez-Cortés**[1], **J. Á. Acosta**[2]

**1** Sección de Mecatrónica, Deparatamento de Ingeniería Eléctrica del Centro de Investigación y de Estudios Avanzados del Instituto Politécnico Nacional, Ciudad de México, México, **2** Departamento de Ingeniería de Sistemas y Automática, Universidad de Sevilla, Sevilla, Spain

☯ These authors contributed equally to this work.
* ytlatelpa@cinvestav.mx

## Abstract

This paper introduces a new decentralized control strategy for an unmanned aerial manipulator (UAM) constrained to the vertical plane. The control strategy comprises two loops: the first compensates for the aerial vehicle's impact on the manipulator; and the second one implements independent controllers for the aerial vehicle and the manipulator. The controller for the aerial vehicle includes an estimator to compensate for the dynamic influence of the manipulator, even if it is affected by external wind-gust disturbances. The manipulator has two revolute joints; however, it is modeled as an dynamically equivalent manipulator, with one revolute and one prismatic joint. The proposed control strategy's performance is evaluated using a simulator that includes the vehicle's aerodynamics and the manipulator's contact force and moment.

## 1 Introduction

Combining multi-rotor Unmanned Aerial Vehicles (UAVs) with robot manipulators has led to the more versatile and agile devices termed Unmanned Aerial Manipulators (UAMs). The applications of UAMs are vast and diverse since UAMs can take advantage of the manipulator's dexterity and the UAV's agility. However, technological and scientific issues must be addressed to exploit their usefulness. Among these problems is the need for lighter materials, better batteries, foolproof safety, and enhanced performance during tracking/positioning manipulation tasks. Synthesizing robust control algorithms can tackle the UAM performance during manipulation tasks.

UAMs are classified by the vehicle's number of rotors, manipulator links, and arms. In particular, the manipulator must have at least one actuated degree of freedom to be considered an UAM. One of the first UAM with a manipulator with 2-DoF revolute joints can be found in [1]. Nowadays, there are many applications including experimental work on UAMs with multiple degrees of freedom.

There are two trends in control architectures reported for UAMs: centralized and decentralized. The first considers the UAM as a whole system, and the second one, accounts for separated dynamic systems, the UAV and the robotic manipulator. In the first approach, only one

**Data Availability Statement:** All relevant data are within the manuscript.

**Funding:** Y.E. Tlatelpa-Osorio 702178 CONSEJO NACIONAL DE CIENCIA Y TECNOLOGÍA https://conahcyt.mx/.

**Competing interests:** The authors have declared that no competing interests exist.

mathematical model describes the UAV and robotic manipulator dynamics. Conversely, in the decentralized approach, the quadrotor and the robotic manipulator are considered separate systems; their interaction is viewed as the effect of mutual disturbances. Sometimes, the control architecture is influenced by the dynamic modeling method used for UAMs. Thus, the centralized approach fits better with the Euler-Lagrange formalism, while the Newton-Euler approach follows the decentralized framework. However, under the same conditions both modeling frameworks are equivalent, generating then the same dynamic models.

As examples of the application within the centralized control approach, in [2] an UAM with two anthropomorfic manipulators is commanded following the impedance control technique; in [3] the dynamic model of the UAM is linearized around an equilibrium point and a Linear Quadratic Regulator (LQR) is employed.

On the other hand, in the decentralized approach, the controllers selected for the UAV and manipulator can have different formulations that may not require knowledge of the whole model. In [4] is employed a kinematic control for the two anthropomorphic UAM's arms, while a backstepping controller is implemented for the hexacopter that carries the arms. In [5], a general description of the robustness of decentralization is provided with nonlinear controller at the kinematic level. In [6], a PID-based controller with gravity compensation is used. In [7], a PD controller is used for the manipulator, while the UAV uses two level controls, one for translational dynamics and one for attitude dynamics, for an ad-hoc type of UAM satisfying differential flatness, i.e. fully linearizable by feedback. The work in [8] uses a passive nonlinear dynamic controller for the UAV and an integral kinematic controller for the manipulator. Adaptive control has also been implemented in [9]. A review of other centralized and descentralized approaches, summarizing characteristics and differences can be found in [10].

Under the decentralized approach, some authors have pointed out the exogenous nature of the disturbances from the robotic manipulator to the aerial vehicle and vice-versa, see e.g. [1] where the effects on the robotic manipulator of the aerial vehicle displacement of the center of mass and changes in the moments of inertia are characterized through experiments. This crucial observation shows that decentralized approaches are more suitable when external disturbances are present, thus allowing to implement robust control strategies for each system independently. In fact, even though many works, such as those described above, have proposed a variety of controllers, no attention have been paid to the actual nature of the robustness gained with the decentralized approaches. This is the main focus of this work where a detailed analysis of the dynamical interaction between the aerial vehicle and the robotic manipulator and the ad-hoc design of a robust controller are provided.

Some efforts have been made to take into account the interaction between the UAV and the robotic manipulator by using equivalent models. For example, in [11], the dynamic model of an n-link manipulator is described by a one-degree-of-freedom (DoF) revolute joint that concentrates the n-link manipulator total mass, assuming that the robot arm reach any commanded reference instantly. Thus, as far as we know, the only work treating theoretically the interaction is in [8], but only at the kinematic level of the robot arm, which means that, in practice, the analysis is only valid for slow movements when the accelerations can be neglected.

The present work addresses the problem of controlling the longitudinal UAM dynamics following the decentralized approach taking into account external disturbances. The longitudinal dynamics considered captures the essential nonlinearities of a 3D environment and most practical aerial missions are 2D immersed in a 3D workspace [8].

The robotic manipulator is composed of two links with revolute joints, where the computed-torque methodology is employed to design a trajectory-tracking controller [12]. The robust control implemented in the UAV is based on the PD methodology in combination with

the estimator of external forces and moments similarly to [13]. As one of the main contributions of this work and to better illustrate that the manipulator effects over the quadrotor can be tailored as exogenous disturbances, the robotic manipulator dynamics are modeled as an equivalent 1-revolute 1-prismatic robotic manipulator, thus the manipulator model approach proposed in [5] is also extended. The Newton-Euler method and its recursive algorithm [14] are used to obtain the UAM dynamic model.

Finally, numerical simulations using the UAM realistic simulator reported in [8] are presented to assess the performances.

The main contributions are summarized as follows:

1. The modelization of a 2-revolute links robotic manipulator as an equivalent 1-revolute, 1-prismatic links robotic manipulator; this permits simplifying the stability analysis, by customizing the dynamic effect of the robotic manipulator on the aerial vehicle as an exogenous disturbance. In the previous authors' works in [8, 11, 15] a simplified model for the n-link manipulator as a 1-revolute joint manipulator was proposed. However, only kinematics is considered for the robot arm interaction and control. The dynamic model developed in this work allows us to characterize its complete nature and formalize the exogenous nature of the dynamic interaction, i.e. torques and forces. We emphasize that this dynamical analysis is missing in the literature, normally neglected with assumptions such as slow motion, that indeed are not practical because there may be external disturbances such as gusts of wind that cause high accelerations.

2. The model developed paves the way for the design of a decentralized robust PD and computed torque nonlinear controllers together with a exogenous disturbances estimator. A complete stability analysis is provided that ensures that the tracking error is confined to a vicinity of the origin exponentially, which is a well-known desirable robust property. The performances are validated through numerical simulations.

The organization of this work is as follows. Section 2 presents the Aerial Manipulator Dynamics model on the plane. In section 3, the control strategy is developed, as well as the stability analysis. In section 4 the numerical simulations results are shown, and conclusions and future work are presented in section 5.

## 2 UAM dynamic model

This work considers a quadrotor with a robotic manipulator at the bottom, i.e an UAM. The robotic manipulator has two degrees of freedom as two revolute joints. The UAM's dynamic model is obtained under the following considerations in all flight operations:

- the aerial vehicle and the robotic manipulator are considered rigid bodies, i.e. links are not flexible;

- the relative motions of the propellers to the quadrotor frame are disregarded;

- the union between the aerial vehicle and the robotic manipulator is rigid and remains unaltered;

- the links move independently only when their actuators generate a moment;

  Recall that the robotic manipulator can only move on the plane $0x^b z^b$, see Fig 1.

  To obtain the UAM dynamic model the reference frames $0x^i y^i z^i$, $i = 1, 2, 3$ on the robotic manipulator's links are defined, as in Fig 1.

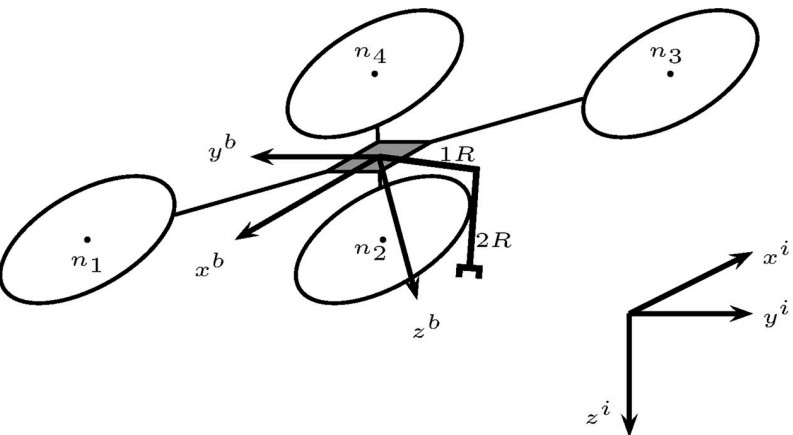

**Fig 1. Unmanned aerial manipulator.** It comprises the aerial vehicle with four rotors $n_1$, $n_2$, $n_3$, and $n_4$; and the robotic manipulator with two revolute joints $R_1$ and $R_2$. Body axes $0x^b y^b z^b$ and inertial axes $0x^i y^i z^i$.

## 2.1 Quadrotor dynamic model

From Newton-Euler laws of motion the quadrotor dynamic model is

$$
\begin{aligned}
m_Q \ddot{X} &= m_Q g e_3 + F_p^i - f_{RM}^i \\
J\dot{\Omega} &= -\Omega \times J\Omega + M_p^b - M_{RM}^b
\end{aligned}
\tag{1}
$$

where $m_Q$ is the quadrotor mass, $g$ is the gravity acceleration constant, $e_3 = [0\ 0\ 1]^\top$, $F_p^i$ is the force due to the propulsion system, $f_{RM}^i$ is the force due to the robotic manipulator expressed in the inertial frame and $X = [x\ y\ z]^\top$ is the vector of cartesian coordinates. Moreover, $J = \text{diag}\{J_{xx}, J_{yy}, J_{zz}\}$ is the quadrotor inertia matrix, $\Omega = [p\ q\ r]^\top$ is the quadrotor velocity in body coordinates, $M_p^b$ the moment due to the propulsion system, and $M_{RM}^b$ the moment due to the robotic manipulator expressed in body coordinates.

The propulsion force is given by

$$
F_p^i = -T_T R e_3
$$

where $T_T = \sum_{i=1}^4 T_i$ is the total thrust produced by the four rotors with $T_i$ the thrust produced by rotor $r_i$. Moreover, $R \in SO(3)$ is the rotation matrix from body coordinates to inertial coordinates. Where $SO(3) = \left\{ R \in \mathbb{R}^{3\times 3} \mid R^\top R = I, \det(R) = 1 \right\}$ with $I$ the identity matrix. Introducing the following notation $c_\sigma = \cos(\sigma)$, $s_\sigma = \sin(\sigma)$ for any angle $\sigma$, the propulsion moment is

$$
M_p^b = \begin{bmatrix} M_x^b \\ M_y^b \\ M_z^b \end{bmatrix} = \begin{bmatrix} \ell c_{\pi/4}(T_2 + T_3 - T_1 - T_4) \\ \ell c_{\pi/4}(T_1 + T_2 - T_3 - T_4) \\ Q_1 + Q_3 - Q_2 - Q_4 \end{bmatrix}
$$

where $\ell$ is the distance between the origin of the body frame and the rotation axis of each rotor, $\pi/4$ is the angle between the rotor arm and the body axis $0x^b$ for rotors 1 and 3; and between the body axis $0y^b$ for rotors 2 y 4. Finally, $Q_i$, $i = 1, \cdots, 4$ is the reaction moment of each rotor.

The force $f_{RM}^i$ and the moment $M_{RM}^b$ can be computed from the inward iteration of the Recursive Newton-Euler Algorithm (RNEA) that propagates the forces and moments from the end effector to the robotic manipulator base. The RNEA procedure is completed with the outward iteration to compute links velocities and acceleration. Both iterations allow to obtain the dynamic model of the robotic manipulator [14].

## 2.2 Robotic manipulator dynamic model

The ideal workspace of the robotic manipulator considered in this work is a semi-circle below the quadrotor. This workspace is achieved through the independent motion of the two revolute joints, $1R$ and $2R$; see Fig 2. Fig 2 also shows the robotic manipulator's center of gravity.

The robotic manipulator motion can be interpreted as the motion of a fully actuated slung load when it is analyzed from the motion of its center of gravity. This observation gave rise to modeling the $1R2R$ robotic manipulator as an equivalent robotic manipulator composed of one revolute joint $1'R$ and one prismatic joint $1P$, as illustrated in Fig 3. This idea was partially developed in [15] by modeling the $1R2R$ robotic manipulator as an actuated pendulum with constant length. Pursuing this idea, in this work, the second degree of freedom is recovered by considering that the pendulum's center of mass can move longitudinally using a prismatic joint, unlike there. As a result, the complete robotic manipulator's workspace can be covered.

The reference systems shown in Fig 3 follow the link-frame procedure proposed in [14]. Moreover, Table 1 summarizes the link parameters, also known as the Denavit-Hartenberg in the proximal variant notation [16]. Fig 4 depicts the link parameters.

From the link parameters, the following rotation matrices are obtained

$$\binom{0}{1}R = \begin{bmatrix} c_{\theta_1} & -s_{\theta_1} & 0 \\ s_{\theta_1} & c_{\theta_1} & 0 \\ 0 & 0 & 1 \end{bmatrix}, \quad \binom{1}{2}R = \begin{bmatrix} 1 & 0 & 0 \\ 0 & 0 & -1 \\ 0 & 1 & 0 \end{bmatrix}$$

$$\binom{2}{3}R = \begin{bmatrix} 1 & 0 & 0 \\ 0 & 1 & 0 \\ 0 & 0 & 1 \end{bmatrix} \tag{2}$$

the rotation matrices use the notation introduced in [14]. Hence, $\binom{i+1}{i}R$ is the rotation matrix from reference frame $i$ to $i + 1$. Moreover, $\binom{i+1}{i}R^\top = \binom{i}{i+1}R$ and $\binom{i+1}{i+2}R\binom{i+2}{i}R = \binom{i+1}{i}R$.

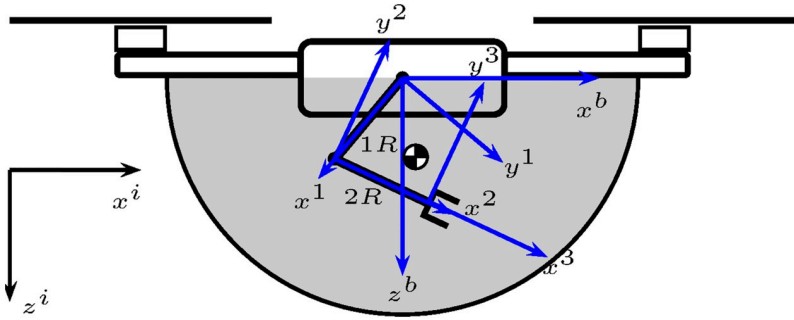

**Fig 2. UAM ideal workspace.**

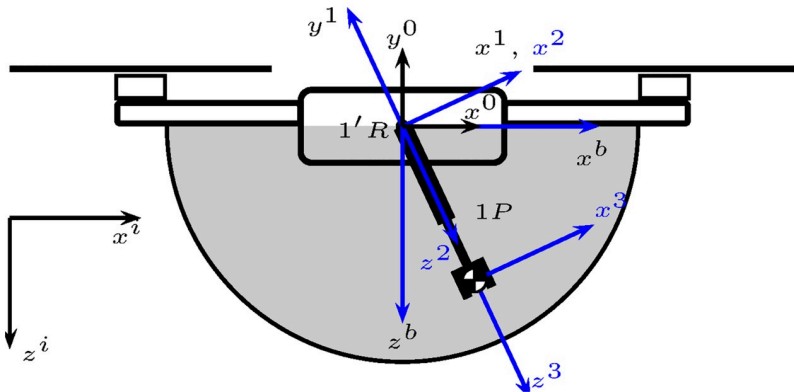

**Fig 3. UAM ideal workspace with revolute and prismatic joints.**

Also, the distance $P^i_{i+1}$ between frame $i$ and frame $i + 1$ measured from frame $i$ is given by

$$P^0_1 = \begin{bmatrix} 0 \\ 0 \\ 0 \end{bmatrix} \quad P^1_2 = \begin{bmatrix} 0 \\ 0 \\ 0 \end{bmatrix} \quad P^2_3 = \begin{bmatrix} 0 \\ 0 \\ L \end{bmatrix} \tag{3}$$

Now, without any loss of generality, the following simplifications are considered to tailor the robotic manipulator dynamic model.

The revolute joint $1'R$ is massless so that the mass of the revolute joints $1R$ and $2R$ is concentrated at the distal end of equivalent prismatic link $P_1$. Hence,

$$m_{1'R} = 0, \quad m_P = m_{1R} + m_{2R}$$

with $m_{1'R}$ the mass of the revolute joint $1'R$, $m_P$ the mass of the revolute joint $1P$ and $m_{1R}$, $m_{2R}$ the mass of the revolute joints $1R$ and $2R$, respectively. Moreover, the position $P^i_C$ of the center of mass of link $i$ expressed in the $i$-th reference frame, is defined by the following vectors

$$P^1_C = \begin{bmatrix} 0 \\ 0 \\ 0 \end{bmatrix}, \quad P^2_C = \begin{bmatrix} 0 \\ 0 \\ L \end{bmatrix} \tag{4}$$

moreover, the inertia tensors for each point mass are $I^1_C = 0$ and $I^2_C = 0$.

**Table 1. Link parameters.**

| $i$ | $\alpha_{i-1}$ | $a_{i-1}$ | $d_i$ | $\theta_i$ |
|-----|----------------|-----------|-------|------------|
| 1 | $\alpha_0 = 0$ | $a_0 = 0$ | $d_1 = 0$ | $\theta_1$ |
| 2 | $\alpha_1 = \frac{\pi}{2}$ | $a_1 = 0$ | $d_2 = 0$ | $\theta_2 = 0$ |
| 3 | $\alpha_2 = 0$ | $a_2 = 0$ | $d_3 = L$ | $\theta_3 = 0$ |

With $\alpha_i$ the angle from $z^i$ to $z^{i+1}$ measured around $x^i$, $a_i$ is the distance from $z^i$ to $z^{i+1}$ measured along $x^i$, $d_i$ is the distance from $x^{i-1}$ to $x^i$ measured along $z^i$, and $\theta_i$ is the angle from $x^{i-1}$ to $x^i$ measured around $z^i$.

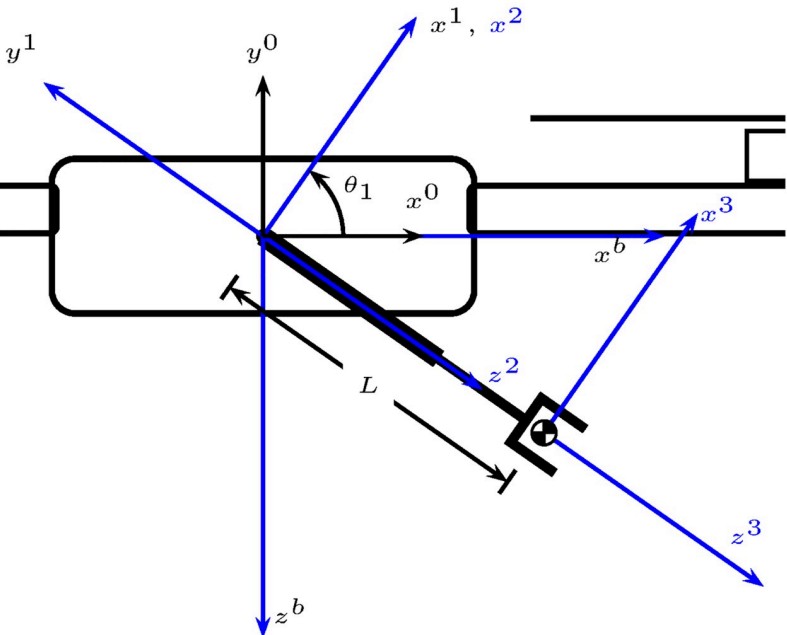

**Fig 4. UAM link variables.**

Since the robotic manipulator is attached to a flying base, it is necessary to obtain the rotation matrix between the reference frame $0x^0y^0z^0$ and the body reference frame $0x^by^bz^b$. Note that both frames are rigidly attached but have different configurations, see Fig 4. The corresponding rotation matrix is the following.

$$
\binom{0}{b}R = \begin{bmatrix} 1 & 0 & 0 \\ 0 & 0 & -1 \\ 0 & 1 & 0 \end{bmatrix}
$$

Thus, the boundary conditions for the flying base are [17–19]

$$
\Omega^0 = \binom{0}{b}R\Omega + \dot{\theta}_0 \hat{z}^0
$$

$$
\dot{\Omega}^0 = \binom{0}{b}R\dot{\Omega} + \binom{0}{b}R\Omega \times \dot{\theta}_0 \hat{z}^0 + \ddot{\theta}_0 \hat{z}^0
$$

$$
\dot{V}^0 = \binom{0}{b}R\left[\dot{V}^b + \Omega \times V^b + \Omega \times P_0^b + \Omega \times (\Omega \times P_0^b) - gR^\top e_3\right]
$$

where $\Omega^0$ and $V^0$ are the angular and translational velocities of the frame $0x^0y^0z^0$, and $V^b = R^\top \dot{X}$ is the translational velocity of the quadrotor expressed in body axes. Moreover, $P_0^b = 0 \in \mathbb{R}^3$ is the distance from the frame $0x^0y^0z^0$ to the body frame and $\theta_0$ is the angle of rotation of frame $0x^0y^0z^0$ with respect to frame $0x^by^bz^b$ around the $\hat{z}^0$ axis. The term $R^\top ge_3$ is introduced to consider the gravitational acceleration. In the following, $\Omega^i, \dot{\Omega}^i, V^i$ and $\dot{V}^i$ are the rotational and translational velocities and accelerations of joint $i$ expressed in the frame $0x^iy^iz^i$, respectively.

The frame $0x^0y^0z^0$ is rigidly attached to the body frame so that $\theta_0 = \dot{\theta}_0 = \ddot{\theta}_0 = 0$, thus the boundary conditions reduce to

$$\Omega^0 = \left({}_b^0 R\right)\Omega$$

$$\dot{\Omega}^0 = \left({}_b^0 R\right)\dot{\Omega}$$

$$\dot{V}^0 = \left({}_b^0 R\right)\left(\dot{V}^b + \Omega \times V^b - gR^\top e_3\right)$$

The angular velocities are propagated to frames $0x^1y^1z^1$ and $0x^2y^2z^2$ following the outward iteration of the RNEA method, see Appendix A, as follows. Therefore, the angular velocity for frame $0x^1y^1z^1$ attached to a revolute joint with $i + 1 = 1$ is given by (44)

$$\Omega^1 = \left({}_0^1 R\right)\Omega^0 + \dot{\theta}_1 \hat{z}^1$$

meanwhile, for frame $0x^2y^2z^2$ attached to the prismatic joint with $i + 1 = 2$, it follows that (45)

$$\Omega^2 = \left({}_1^2 R\right)\Omega^1$$

Once again, from the RNEA outward iteration, the angular and translational accelerations are propagated as follows. For the frames $0x^1y^1z^1$ and $0x^2y^2z^2$ with $i + 1 = 1$ and $i + 1 = 2$, respectively, one obtains (46) and (47),

$$\dot{\Omega}^1 = \left({}_0^1 R\right)\dot{\Omega}^0 + \left({}_0^1 R\right)\Omega^0 \times \dot{\theta}_1 \hat{z}^1 + \ddot{\theta}_1 \hat{z}^1$$

$$\dot{V}^1 = \left({}_0^1 R\right)\left[\dot{\Omega}^0 \times P_1^0 + \Omega^0 \times (\Omega^0 \times P_1^0) + \dot{V}^0\right]$$

For the prismatic joint, one has (48) and (49),

$$\dot{\Omega}^2 = \left({}_1^2 R\right)\dot{\Omega}^1$$

$$\dot{V}^2 = \left({}_1^2 R\right)\left[\dot{\Omega}^1 \times P_2^1 + \Omega^1 \times (\Omega^1 \times P_2^1) + \dot{V}^1\right] + 2\Omega^2 \times \dot{L}\hat{z}^2 + \ddot{L}\hat{z}^2.$$

Taking into account $P_1^0$ and $P_2^1$ defined in Eq (3) the accelerations $\dot{V}^1$ and $\dot{V}^2$ reduce to

$$\dot{V}^1 = \left({}_0^1 R\right)\dot{V}^0$$

$$\dot{V}^2 = \left({}_1^2 R\right)\dot{V}^1 + 2\Omega^2 \times \dot{L}\hat{z}^2 + \ddot{L}\hat{z}^2$$

The last step of the outward iteration involves the computation of the links' center of mass acceleration, forces and moments acting on it. The acceleration of the link's center of mass is computed as follows (50)

$$\dot{V}_C^1 = \dot{\Omega}^1 \times P_C^1 + \Omega^1 \times (\Omega^1 \times P_C^1) + \dot{V}^1$$

$$\dot{V}_C^2 = \dot{\Omega}^2 \times P_C^2 + \Omega^2 \times (\Omega^2 \times P_C^2) + \dot{V}^2$$

Considering (4) the acceleration of the revolute link center of mass reduces to

$$\dot{V}_C^1 = \dot{V}^1$$

The forces acting at the center of mass of each link are (51)

$$F^1 = m_{R_1'}\dot{V}^1 = 0$$

$$F^2 = m_{P_1}\dot{V}_C^2$$

Finally, under the aforementioned considerations, the moments on each link are $N^1 = N^2 = 0$, and hence the outward iteration is completed.

The inward iteration propagates forces and moments acting on the end effector to the robotic manipulator base. The inward algorithm runs from $i + 1 = 3$ to $i + 1 = 2$. Thus, the force $f^2$ exerted on link 2, by link 1 and the force $f^1$ exerted on link 1 by the robotic manipulator's base are (52)

$$\begin{aligned} f^2 &= F^2 \\ f^1 &= ({}_2^1R)f^2 \end{aligned}$$

(5)

where it is assumed that $f^3 = 0$. Additionally, the torque $n^2$ exerted on link 2 by link 1 and the torque $n^1$ exerted on link 1 by the robotic manipulator base are given by

$$\begin{aligned} n^2 &= P_C^2 \times F^2 \\ n^1 &= ({}_2^1R)n^2 + P_2^1 \times ({}_2^1R)f^2 \end{aligned}$$

(6)

where it is assumed that $n^3 = 0$. Considering (3) and (4), it follows that

$$n^1 = ({}_2^1R)n^2$$

Finally, the robotic manipulator dynamic model is described by the following equations

$$\begin{aligned} \tau_R &= n^{1\top}\hat{z}^1 \\ f_P &= f^{2\top}\hat{z}^2 \end{aligned}$$

(7)

where $\tau_P$ is the control moment applied to the revolute link, and $f_P$ is the control force applied to the prismatic link. The force $f^1$ and the moment $n^1$ can be expressed in the reference frame $0x^0y^00z^0$ as follows

$$\begin{aligned} f^0 &= ({}_1^0R)f^1 \\ n^0 &= ({}_1^0R)n^1 \end{aligned}$$

(8)

thus,

$$\begin{aligned} f_{RM}^i &= R({}_b^0R)^\top f^0 \\ n_{RM}^b &= ({}_b^0R)^\top n^0 \end{aligned}$$

(9)

In the following the angle $\theta_1$ is replaced by $\theta_P$ measured as shown in Fig 5. Thus,

$$\theta_1 = \theta_P - \theta_Q$$

Complex but straightforward computations show that the quadrotor translational dynamic model constrained to the plane $0x^iz^i$ becomes

$$\bar{M}\begin{bmatrix} \ddot{x} \\ \ddot{z} \end{bmatrix} = \bar{M}g\begin{bmatrix} 0 \\ 1 \end{bmatrix} - T_T\begin{bmatrix} s_{\theta_Q} \\ c_{\theta_Q} \end{bmatrix} - \begin{bmatrix} \bar{f}_{RM_x}^i \\ \bar{f}_{RM_z}^i \end{bmatrix}$$

(10)

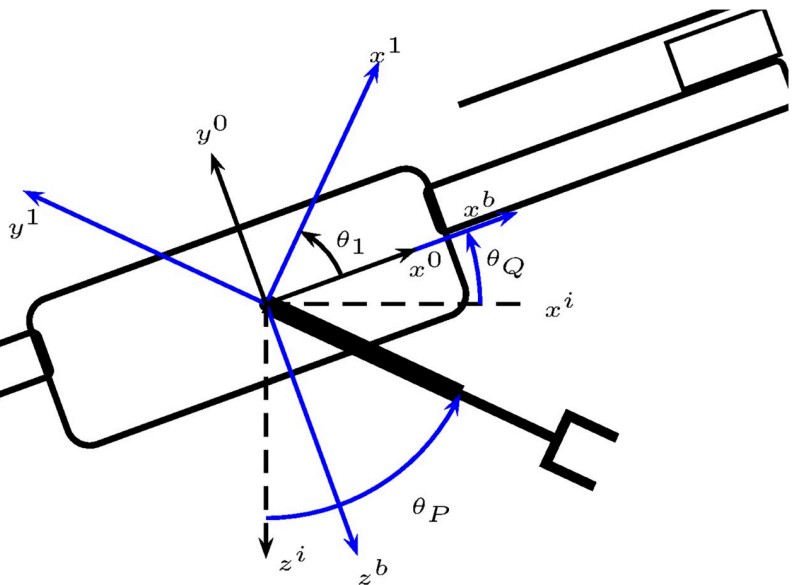

**Fig 5. Prismatic link angle with respect to the inertial frame.**

where the following identity has been considered $R^\top \dot{X} = \dot{V}^b + \Omega \times V^b$, with $\bar{M} = m_Q + m_{P_1}$ and

$$
\begin{aligned}
\bar{f}_{RM}^i &= \begin{bmatrix} \bar{f}_{RM_x}^i \\[4pt] \bar{f}_{RM_y}^i \\[4pt] \bar{f}_{RM_z}^i \end{bmatrix} \\[10pt]
&= m_{P_1} \begin{bmatrix} c_{\theta_P}(L\ddot{\theta}_P + 2\dot{L}\dot{\theta}_P) + s_{\theta_P}(\ddot{L} - L\dot{\theta}_P^2) \\[4pt] 0 \\[4pt] s_{\theta_P}(L\ddot{\theta}_P + 2\dot{L}\dot{\theta}_P) + c_{\theta_P}(L\dot{\theta}_P^2 - \ddot{L}) \end{bmatrix}
\end{aligned}
$$

The rotational dynamics constrained to the $0x^i z^i$ plane becomes

$$
J_{yy}\ddot{\theta}_Q = M_y^b - n_{RM_q}^b \tag{11}
$$

where

$$
n_{RM}^b = \begin{bmatrix} n_{RM_p}^b \\[4pt] n_{RM_q}^b \\[4pt] n_{RM_r}^b \end{bmatrix}
$$

with $n_{RM_p}^b = n_{RM_r}^b = 0$ and the following identities are instrumental

$$
c_{(\theta_P - \theta_Q)}c_{\theta_Q} - s_{(\theta_P - \theta_Q)}s_{\theta_Q} = c_{\theta_P}, \quad s_{(\theta_P - \theta_Q)}c_{\theta_Q} + c_{(\theta_P - \theta_Q)}s_{\theta_Q} = s_{\theta_P}
$$

$$
\begin{aligned}
n_{RM_q}^b &= m_P L^2 \ddot{\theta}_P + 2m_P L\dot{L}\dot{\theta}_P + m_P g L s_{(\theta_P - \theta_Q)}c_{\theta_Q} - \frac{m_P}{M}Lc_{(\theta_P - \theta_Q)}c_{\theta_Q}\bar{f}_{RM_x} \\[4pt]
&\quad + \frac{m_P}{M}Lc_{(\theta_P - \theta_Q)}s_{\theta_Q}\bar{f}_{RM_z}
\end{aligned}
$$

Finally, from (7) the robotic manipulator dynamic becomes

$$
\begin{bmatrix} m_P L^2 & 0 \\ 0 & m_P \end{bmatrix} \begin{bmatrix} \ddot{\theta}_P \\ \ddot{L} \end{bmatrix} +
$$

$$
\begin{bmatrix} m_P L \dot{L} & m_P L \dot{\theta}_P \\ -m_P L \dot{\theta}_P & 0 \end{bmatrix} \begin{bmatrix} \dot{\theta}_P \\ \dot{L} \end{bmatrix} +
$$

$$
\begin{bmatrix} m_P L g s_{(\theta_P - \theta_Q)} c_{\theta_Q} \\ -m_P g s_{(\theta_P - \theta_Q)} s_{\theta_Q} \end{bmatrix} +
$$

$$
\begin{bmatrix} -\frac{1}{M} c_{(\theta_P - \theta_Q)} c_{\theta_Q} & \frac{1}{M} c_{(\theta_P - \theta_Q)} s_{\theta_Q} \\ -\frac{m_P}{M} s_{(\theta_P - \theta_Q)} c_{\theta_Q} & \frac{m_P}{M} s_{(\theta_P - \theta_Q)} s_{\theta_Q} \end{bmatrix} \times \begin{bmatrix} \bar{f}^i_{RM_x} \\ \bar{f}^i_{RM_z} \end{bmatrix} = \begin{bmatrix} \tau_R \\ f_P \end{bmatrix}
$$

(12)

Summarizing, the UAM dynamic model that will be considered for control design is described by Eqs (10), (11) and (12).

## 3 Decentralized robust control strategy

The control design is divided into two control loops: the inner loop is a state feedback controller that customizes the robotic manipulator effects on the aerial vehicle as an exogenous disturbance (C4D), generating a decentralized UAM dynamic model. In contrast, the outer control loop uses the decentralized model and independently applies control strategies for the quadrotor and robotic arm. We first present the C4D state feedback and then the estimator of the quadrotor exogeneous moments and forces (EQEMF).

Since the quadrotor acts as a flying base for the manipulator, the quadrotor and the robotic manipulator dynamics are interconnected. Hence, assuming that the forces and moments from the inward iteration are exogenous signals is not trivial, being this development and analysis contributions of the present work. Therefore, the first step is to show that the manipulator dynamics can be considered exogenous disturbances for the quadrotor dynamics and vice versa. Since the robotic manipulator dynamic is fully actuated, its interaction with the quadrotor can be characterized as exogenous, thus allowing to design a control scheme to compensate for its effects.

The following controller is proposed to make the disturbances on the quadrotor from the manipulator exogenous and vice versa.

$$
\begin{aligned}
M^b_y = {} & m_P g L c_{(\theta_P - \theta_Q)} s_{\theta_Q} + \frac{m_P}{M} L s_{(\theta_P - \theta_Q)} s_{\theta_Q} \bar{f}_{RM_x} \\
& + \frac{m_P}{M} L s_{(\theta_P - \theta_Q)} c_{\theta_Q} \bar{f}_{RM_z} + \bar{M}^y_b
\end{aligned}
$$

(13)

$$\tau_R = -\frac{1}{M}\left(c_{(\theta_P-\theta_Q)}c_{\theta_Q}\bar{f}^i_{RM_x} - c_{(\theta_P-\theta_Q)}s_{\theta_Q}\bar{f}^i_{RM_z}\right)$$
$$+ m_P Lg c_{(\theta_P-\theta_Q)}s_{\theta_Q} + \bar{\tau}_R \tag{14}$$

$$f = -\frac{m_P}{M}\left(s_{(\theta_P-\theta_Q)}c_{\theta_Q}\bar{f}^i_{RM_x} - s_{(\theta_P-\theta_Q)}s_{\theta_Q}\bar{f}^i_{RM_z}\right)$$
$$+ m_P g c_{(\theta_P-\theta_Q)}c_{\theta_Q} + \bar{f}_P \tag{15}$$

The UAM dynamics (10)–(12) with the inner loop controller RC4D (13)–(15) results in

$$\bar{M}\ddot{\mathbf{X}} = \bar{M}g\mathbf{e} - T_T\mathbf{r}_{\theta_Q} + \delta_T$$

$$\mathbf{J_M}\begin{bmatrix}\ddot{\theta}_Q \\ \ddot{\theta}_P \\ \ddot{L}\end{bmatrix} = \begin{bmatrix}\delta_R \\ -2m_P L\dot{L}\dot{\theta}_P - m_P Lg s_{\theta_P} \\ -m_P L\dot{\theta}_P^2 + m_P g c_{\theta_P}\end{bmatrix} + \mathbf{I_{3\times3}}\begin{bmatrix}\bar{M}^y_b \\ \bar{\tau}_R \\ \bar{f}_P\end{bmatrix} \tag{16}$$

where

$$\mathbf{X} = \begin{bmatrix}x \\ z\end{bmatrix}, \ \mathbf{e} = \begin{bmatrix}0 \\ 1\end{bmatrix}, \ \mathbf{r}_{\theta_Q} = \begin{bmatrix}s_{\theta_Q} \\ c_{\theta_Q}\end{bmatrix}, \ \delta_T = \begin{bmatrix}\bar{f}^i_{RM_x} \\ \bar{f}^i_{RM_z}\end{bmatrix},$$

$$\mathbf{J_M} = \begin{bmatrix}J_{yy} & 0 & 0 \\ 0 & m_P L^2 & 0 \\ 0 & 0 & m_P\end{bmatrix}, \ \mathbf{I_{3\times3}} = \begin{bmatrix}1 & 0 & 0 \\ 0 & 1 & 0 \\ 0 & 0 & 1\end{bmatrix}$$

and

$$\delta_R = m_P L^2\ddot{\theta}_P + 2m_P L\dot{L}\dot{\theta}_P + m_P gL s_{\theta_P}.$$

**Remark 1** *It is important to underscore, that the combination of both the coordinate change and the controller is what ensures that $\delta_T$ and $\delta_R$ can be treated as external disturbances for the quadrotor dynamics comming from the robotic manipulator. Furthermore, the proposed change of coordinates paved the way to the such controller design. More importantly, the proposed design allows the online estimation of the signals $\delta_T$, $\delta_R$, as it is described in the following developments.*

### 3.1 Quadrotor exogeneous disturbances estimator

The estimator of external forces and moments for a quadrotor was introduced in [13]. This estimator was also employed in the previous version of this study in [15]. The estimator is based on the Immersion and Invariance method [20]. As it will be evident in the following developments, if the disturbance is not exogenous, the estimator assumptions fail to be fulfilled.

First, the quadrotor dynamics is rewritten as

$$\dot{\zeta}_1 = \bar{f}_1(\zeta_1, \mathbf{r}_{\theta_Q}, T_T) + \delta_T$$
$$\dot{\zeta}_2 = \bar{f}_2(\bar{M}^y_b) + \delta_R$$

with $\zeta_1 = \bar{M}\dot{\mathbf{X}}$, $\zeta_2 = J_{yy}\dot{\theta}_Q$,

$$\begin{aligned} \bar{f}_1 &= \bar{M}g^e e_3 - T_T \mathbf{r}_{\theta_Q}{}^b e_3 \\ \bar{f}_2 &= \bar{M}_b^y \end{aligned}$$

The external forces and moments errors are defined as

$$\begin{aligned} \widetilde{\delta}_1 &= \delta_T - \eta_1 + \beta_1(\zeta_1) \\ \widetilde{\delta}_2 &= \dot{\delta}_T - \eta_2 + \beta_2(\zeta_1) \\ \widetilde{\delta}_3 &= \ddot{\delta}_T - \eta_3 + \beta_3(\zeta_1) \\ \widetilde{\delta}_4 &= \delta_R - \eta_4 + \beta_4(\zeta_2) \end{aligned} \tag{17}$$

where $\eta_i$, $i = 1, 2, 3, 4$ are the estimator states $\beta_i(\zeta_1)$, $i = 1, 2, 3, \beta_4(\zeta_2)$ are functions defined on the design process.

Note that $\lim_{t\to\infty} \widetilde{\delta}_i = 0$, $i = 1, 2, 3$, implies that the following relationships hold

$$\begin{aligned} \lim_{t\to\infty} \eta_1 - \beta_1(\zeta_1) &= \delta_T, \quad \lim_{t\to\infty} \eta_2 - \beta_2(\zeta_1) = \dot{\delta}_T \\ \lim_{t\to\infty} \eta_3 - \beta_3(\zeta_1) &= \ddot{\delta}_T, \quad \lim_{t\to\infty} \eta_4 - \beta_4(\zeta_2) = \delta_R \end{aligned}$$

Defining the dynamics of the estimator states as follow

$$\begin{aligned} \dot{\eta}_1 &= \eta_2 - \beta_2(\zeta_1) + \frac{\partial \beta_1}{\partial \zeta_1}\left[\bar{f}_1 + \eta_1 - \beta_1(\zeta_1)\right] \\ \dot{\eta}_2 &= \eta_3 - \beta_3(\zeta_1) + \frac{\partial \beta_2}{\partial \zeta_1}\left[\bar{f}_1 + \eta_1 - \beta_1(\zeta_1)\right] \\ \dot{\eta}_3 &= \frac{\partial \beta_3}{\partial \zeta_1}\left[\bar{f}_1 + \eta_1 - \beta_1(\zeta_1)\right] \end{aligned}$$

and choosing $\beta_i(\zeta)$, $i = 1, 2, 3$ such as

$$\frac{\partial \beta_1}{\partial \zeta_1} = -\Gamma_1, \; \frac{\partial \beta_2}{\partial \zeta_1} = -\Gamma_2, \; \frac{\partial \beta_3}{\partial \zeta_1} = -\Gamma_3, \tag{18}$$

with $\Gamma_i$, $i = 1, 2, 3$ positive definite matrices, the external forces estimator dynamics become

$$\widetilde{\delta}_1^{(3)} + \Gamma_1\ddot{\widetilde{\delta}}_1 + \Gamma_2\dot{\widetilde{\delta}}_1 + \Gamma_3\widetilde{\delta}_1 = \delta_T^{(3)} \tag{19}$$

Following the same procedure for the external moments estimator, its dynamics results in

$$\dot{\widetilde{\delta}}_4 + \Gamma_4\widetilde{\delta}_4 = \dot{\delta}_R \tag{20}$$

with

$$\dot{\eta}_4 = \frac{\partial \beta_4}{\partial \zeta_2}\left[\bar{f}_2 + \eta_4 - \beta_4(\zeta_2)\right]$$

and

$$\frac{\partial \beta_4}{\partial \zeta_2} = -\Gamma_4 \tag{21}$$

being $\Gamma_4$ a positive defined matrix.

## 3.2 Quadrotor position and attitude control

The control design for the quadrotor position starts by defining the trajectory tracking error as

$$\widetilde{\mathbf{X}} = \mathbf{X} - \mathbf{X}_d$$

where $\mathbf{X}_d$ is the reference position. Then we have

$$\ddot{\widetilde{\mathbf{X}}} = g\mathbf{e} - \frac{T_T}{\bar{M}}\mathbf{r}_{\theta_Q} + \frac{\delta_T(t)}{\bar{M}} - \ddot{\mathbf{X}}_d \tag{22}$$

The vertical dynamics are directly controlled with $T_T$, meanwhile the horizontal dynamics on the axis $0x^i$ are underactuated and controlled by modifying $\theta_Q$.

First, we rewrite the term $T_T\mathbf{r}_{\theta_Q}$ as follows

$$\frac{T_T}{\bar{M}}\mathbf{r}_{\theta_Q} = \frac{T_T}{\bar{M}\mathbf{r}_{\theta_{Q_d}}^\top\mathbf{r}_{\theta_Q}}\left[\left(\mathbf{r}_{\theta_{Q_d}}^\top\mathbf{r}_{\theta_Q}\right)\mathbf{r}_{\theta_Q}\right]$$

where $\mathbf{r}_{\theta_{Q_d}}$ the desired value for $\mathbf{r}_{\theta_Q}$. Now, the following term is added and subtracted

$$\frac{1}{\bar{M}}\frac{T_T}{\mathbf{r}_{\theta_{Q_d}}^\top\mathbf{r}_{\theta_Q}}\mathbf{r}_{\theta_{Q_d}}.$$

and hence (22) becomes

$$\ddot{\widetilde{\mathbf{X}}} = g\mathbf{e}_2 + \frac{\delta_T}{\bar{M}} - \ddot{\mathbf{X}}_d - \frac{1}{\bar{M}}\frac{T_T}{\mathbf{r}_{\theta_{Q_d}}^\top\mathbf{r}_{\theta_Q}}\mathbf{r}_{\theta_{Q_d}} - \frac{1}{\bar{M}}\Theta$$

The term $\Theta$ is defined as

$$\Theta = \frac{T_T}{\mathbf{r}_{\theta_{Q_d}}^\top\mathbf{r}_{\theta_Q}}\left[\left(\mathbf{r}_{\theta_{Q_d}}^\top\mathbf{r}_{\theta_Q}\right)\mathbf{r}_{\theta_Q} - \mathbf{r}_{\theta_{Q_d}}\right]$$

Let us define the control input $T_T$ and $\mathbf{r}_{\theta_{Q_d}}$ in the following form

$$T_T = u^\top\mathbf{r}_{\theta_Q}, \quad \mathbf{r}_{\theta_{Q_d}} = \frac{u}{\|u\|} \tag{23}$$

where $u = [u_x\ u_z]^\top$ is a new control input. The final controller is defined through $u$ as follows

$$u = \bar{M}\left(K_{PX}\widetilde{\mathbf{X}} + K_{DX}\dot{\widetilde{\mathbf{X}}} + g\mathbf{e}_3 + \ddot{\mathbf{X}}_d\right) + \eta_1 - \beta_1 \tag{24}$$

with $\eta_1 - \beta_1$ the exogenous estimation (19) of the disturbance. The closed loop dynamics results in

$$\ddot{\widetilde{\mathbf{X}}} = -K_{PX}\widetilde{\mathbf{X}} - K_{DX}\dot{\widetilde{\mathbf{X}}} + \frac{\widetilde{\delta}_1}{\bar{M}} - \frac{1}{\bar{M}}\Theta$$

where $K_{PX}$ and $K_{DX}$ are positive definite gain matrices.

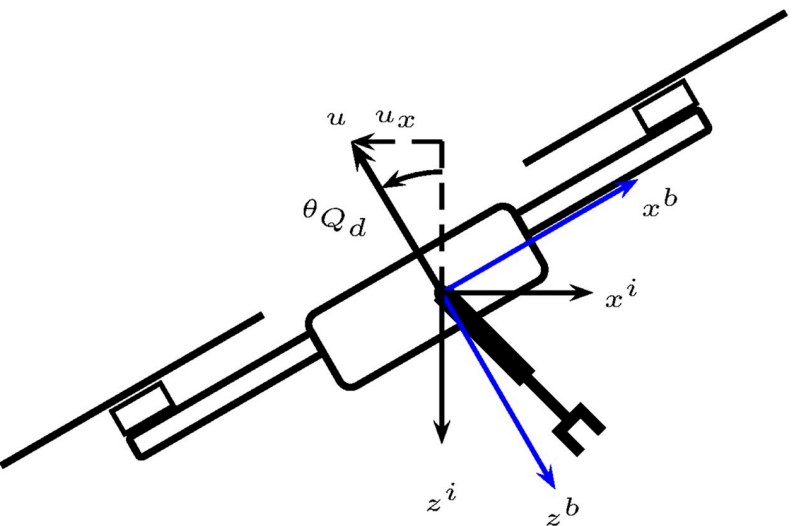

**Fig 6. The angle $\theta_{Q_d}$.**

Due to the underactuated nature of the, translational dynamics, the desired angle $\theta_{Qd}$ is defined geometrically in Fig 6, where $u_x$ is the component on the direction of the $0x^i$ axis of the control vector $u$, thus, one has

$$\theta_{Qd} = \arcsin\left(\frac{u_x}{\|u\|}\right) \tag{25}$$

Then, the quadrotor attitude control input can then be defined as

$$\bar{M}_b^y = J_{yy}(-K_{PQ}(\theta_Q - \theta_{Q_d}) - K_{DQ}(\dot{\theta}_Q - \dot{\theta}_{Q_d}) + \ddot{\theta}_{Qd}) - (\eta_4 - \beta_4) \tag{26}$$

where $K_{PQ}$ and $K_{DQ}$ are positive gains and $\eta_4 - \beta_4$ is the exogenous estimation (20) of the disturbance on the attitude dynamics.

### 3.3 Robot arm controller based on the equivalent model

The controller design is completed with the following control inputs for the revolute and the prismatic joints, for the equivalent manipulator.

$$\bar{\tau}_R = 2m_P L\dot{L}\dot{\theta}_P + m_P Lgs_{\theta_P}$$
$$-m_P L^2\left(K_{PA}(\theta_P - \theta_{P_d}) + K_{DA}(\dot{\theta}_P - \dot{\theta}_{P_d}) - \ddot{\theta}_{Pd}\right) \tag{27}$$

$$\bar{f}_P = m_P L\dot{\theta}_P^2 + m_P gc_{\theta_P}$$
$$-m_P\left(K_{PP}(L - L_d) + K_{DP}(\dot{L} - \dot{L}_d) - \ddot{L}_d\right) \tag{28}$$

where $K_{PA}$, $K_{DA}$, $K_{PP}$ and $K_{DP}$ are positive gains, and $\theta_{P_d}$ and $L_d$ the desired trajectories for $\theta_P$ and $L$, respectively.

### 3.4 UAM closed loop dynamics

To sum up everything, the controller and estimator proposed provide the following UAM closed loop dynamics

$$
\begin{aligned}
\dot{\widetilde{X}}_1 =&\ \widetilde{X}_2 \\[4pt]
\dot{\widetilde{X}}_2 =&\ -K_{PX}\widetilde{X}_1 - K_{DX}\widetilde{X}_2 \\[4pt]
& -\frac{u^\top \mathbf{r}_{\theta_Q}}{m\cos(\widetilde{\theta}_{Q1})}\left\{\cos(\widetilde{\theta}_{Q1})\begin{bmatrix}\cos(\widetilde{\theta}_{Q1}+\theta_{Q_d})\\\sin(\widetilde{\theta}_{Q1}+\theta_{Q_d})\end{bmatrix}-\begin{bmatrix}\cos(\theta_{Q_d})\\-\sin(\theta_{Q_d})\end{bmatrix}\right\}+\frac{1}{m}\mu_1\chi \\[4pt]
\dot{\widetilde{\theta}}_{Q1} =&\ \widetilde{\theta}_{Q2}-\frac{u^\top S[(K_{DX}-\Gamma_1)\mu_1+(K_{DX}-I)\mu_2]\chi}{u^\top u} \\[4pt]
\dot{\widetilde{\theta}}_{Q2} =&\ -K_{PQ}\widetilde{\theta}_{Q1}-K_{DQ}\widetilde{\theta}_{Q2}-F(u,\chi,\widetilde{\delta}_4) \\[4pt]
\dot{\widetilde{\theta}}_{P1} =&\ \widetilde{\theta}_{P2} \\[4pt]
\dot{\widetilde{\theta}}_{P2} =&\ -K_{PA}\widetilde{\theta}_{P1}-K_{DA}\widetilde{\theta}_{P2} \\[4pt]
\dot{\widetilde{L}}_1 =&\ \widetilde{L}_2 \\[4pt]
\dot{\widetilde{L}}_2 =&\ -K_{PP}\widetilde{L}_1-K_{DP}\widetilde{L}_2 \\[4pt]
\dot{\chi} =&\ A_\chi\chi+\mu_3^\top\delta_T^{(3)} \\[4pt]
\dot{\widetilde{\delta}}_4 =&\ -\Gamma_4\widetilde{\delta}_4+\dot{\delta}_R
\end{aligned}
\tag{29}
$$

where $\widetilde{X}_1 = \widetilde{\mathbf{X}}, \widetilde{X}_2 = \dot{\widetilde{\mathbf{X}}}, \theta_{Q1} = \theta_Q - \theta_{Q_d}, \theta_{Q2} = \dot{\theta}_Q - \dot{\theta}_{Q_d}, \theta_{P1} = \widetilde{\theta}_P, \theta_{P2} = \dot{\widetilde{\theta}}_P, \widetilde{L}_1 = \widetilde{L}, \widetilde{L}_2 = \dot{\widetilde{L}},$
$\chi_1 = \widetilde{\delta}_1, \chi_2 = \dot{\widetilde{\delta}}_1, \chi_3 = \ddot{\widetilde{\delta}}_1,$ and

$$
A_\chi = \begin{bmatrix}
-\Gamma_1 & I & 0 & 0 \\
-\Gamma_2 & -\Gamma_1 & I & 0 \\
-\Gamma_3 & -\Gamma_2 & -\Gamma_1 & 0 \\
0 & 0 & 0 & -\Gamma_4
\end{bmatrix},
$$

moreover,

$$
\begin{aligned}
F(u,\chi,\widetilde{\delta}_4) \quad =&\ \frac{1}{(u^\top u)^2}\left(\|u\|^2 u^\top S[(\bar{K}_D+\Gamma 2)\mu_1+\right. \\[4pt]
& (I+\Gamma_1)\mu_2+2\mu_3]\chi- \\[4pt]
& \{2u^\top[(K_D-\Gamma_1)\mu_1+(K_D-I)\mu_2]\chi \\[4pt]
& \left. u^\top S[(K_D-\Gamma_1)\mu_1+(K_D-I)\mu_2]\chi\}\right)-\widetilde{\delta}_4
\end{aligned}
$$

where

$$\begin{aligned}
\mu_1 &= [I_{3\times3}\ 0_{3\times3}\ 0_{3\times3}] \\
\mu_2 &= [0_{3\times3}\ I_{3\times3}\ 0_{3\times3}] \\
\mu_3 &= [0_{3\times3}\ 0_{3\times3}\ I_{3\times3}]
\end{aligned}$$

with $I_{3\times3}$ and $0_{3\times3}$, the identity and zero matrix in $\mathbb{R}^{3\times3}$, respectively.

Function $F(u, \chi, \widetilde{\delta}_4)$ accounts for the estimation error terms that cannot be canceled. Such estimation errors appear because of the control action propagation from the quadrotor rotational dynamics to the quadrotor dynamics along the $0x^i$ axis, this is, through the $\theta_{Q_d}$ computed as

$$\dot\theta_{Qd} = \frac{u^\top S\dot u}{\|u\|^2} \tag{30}$$

$$\ddot\theta_{Qd} = \frac{(u^\top S\ddot u)u^\top u - 2(u^\top \dot u)u^\top S\dot u}{(u^\top u)^2} \tag{31}$$

with

$$\dot u = \bar M\left(K_P\widetilde X_2 + K_D\widetilde X_3\right) + \eta_2 - \beta_2 + K_D\widetilde\delta_1 - \dot{\widetilde\delta}_1 \tag{32}$$

$$\ddot u = \bar M\left(-\bar K_P\widetilde X_2 - \bar K_D\widetilde X_3\right) + \eta_3 - \beta_3 + \bar K_D\widetilde\delta_1 + \dot{\widetilde\delta}_1 + \ddot{\widetilde\delta}_1 + \widetilde\delta_3 \tag{33}$$

where $\widetilde X_3 = -K_P\widetilde X_1 - K_D\widetilde X_2$, $\bar K_P = K_D K_P$ and $\bar K_D = K_P + K_D^2$.

## 3.5 Stability analysis

For the main stability result the following standard assumption for the disturbances is in order.

**Assumption 1** $\delta_T(t)$ and $\delta_T^{(i)}(t)$, $i = 1, 2, 3$ and $\delta_R(t)$ and $\dot\delta_R(t)$ are in $\mathcal{L}_\infty$, $t \geq 0$.

Let us define $x := \mathrm{col}(\widetilde X_1, \widetilde X_2)$, $\widetilde\theta_Q := \mathrm{col}(\widetilde\theta_{Q1}, \widetilde\theta_{Q2})$, $\chi_a := \mathrm{col}(\chi, \widetilde\delta_4)$ and $\Delta_\delta := \mathrm{col}$ $(\mu_3^\top\delta_T^{(3)}, \dot\delta_M)$. Thus, from (29), the $\widetilde x$, $\widetilde\theta_Q$ and $\chi_a$ dynamics in compact form become

$$\begin{aligned}
\dot{\widetilde x} &= A_x\widetilde x + \Psi_x(\widetilde\theta_Q)\cdot\frac{u^\top \mathbf{r}_{\theta_Q}}{m} + D_x\chi_a \\
\dot{\widetilde\theta}_Q &= A_Q\widetilde\theta_Q + \Psi_Q(u)\cdot\chi_a \\
\dot\chi_a &= A_{\chi_a}\chi_a + \Delta_\delta
\end{aligned} \tag{34}$$

where $\Psi_{(\cdot)}$ and $\Delta_\delta$ are vector functions and $D_x$ a constant matrix of appropriate dimensions, respectively, whose expressions can be easily obtained by direct substitutions in (29). Additionally, Hurwitz matrices are defined as

$$A_x := \begin{bmatrix} 0 & I_2 \\ -K_P & -K_D \end{bmatrix}; \quad A_Q := \begin{bmatrix} 0 & I_2 \\ -K_{PR} & -K_{DR} \end{bmatrix};$$

where $K_P$, $K_D$, $K_{PR}$ and $K_{PD}$ are positive definite matrices and let $A_{\chi_a} := \mathrm{Blockdiag}(A_\chi, -\Gamma_4)$. Thus, the main stability result is stated in the following proposition.

**Proposition 1** *Consider the closed-loop dynamics* (29) *with all the control-gain matrices positive definite. Then, under Assumption 1, the error is ultimately bounded to a $\Delta_\delta$-vicinity of the origin. Moreover, if $\Delta_\delta(t) = 0$ then the error converges exponentially to zero, $t \geq 0$.*

**Proof 1** *First, from* (29) *is straightforward to see that the error dynamics $\widetilde{L}_i$ and $\widehat{\theta}_{Pi}$, $i = 1, 2$, are decoupled from others and hence converge exponentially to zero. Therefore, we only focus on the remaining error dynamics* (34). *For, let us define the error $\mathbf{e} := \mathrm{col}(\widetilde{x}, \widetilde{\theta}_Q, \chi_a)$ such that* (34) *can be rewritten in matrix form as*

$$\dot{\mathbf{e}} = \mathcal{A}\mathbf{e} + \begin{bmatrix} 0 \\ 0 \\ \Delta_\delta \end{bmatrix}, \mathcal{A} := \begin{bmatrix} A_x & \Delta_x \dfrac{u^\top \mathbf{r}_{\theta_Q}}{m} & D_x \\ 0 & A_Q & \Psi_Q \\ 0 & 0 & A_{\chi_a} \end{bmatrix}, \tag{35}$$

*where we have omitted all the arguments for compactness. Notice that, by design $\Psi_x(0) = 0$ and hence the whole term can be factored by $\Delta_x$ as in* (35). *The results follow noting that the block-triangular matrix $\mathcal{A}$ is Hurwitz and $\Delta_\delta$ is uniformly bounded under Assumption 1.*

## 4 Numerical simulations

Numerical simulations were driven on the UAM realistic simulator reported in [8] and were kindly provided to us by Carlos Rodríguez de Cos from the University of Sevilla. The original simulation platform consists of 4 main blocks: the MANT mathematical model block, the target trajectory block for the UAV and the End effector, a block for the UAV control, and one for the manipulator control. The MANT system is disturbed by a random gust of wind.

Note that the proposed controller was designed considering a quadrotor with a manipulator composed of a revolute joint R1 and a prismatic joint P1; however, the simulator considers only revolute joints. Hence, it is mandatory to prove that both manipulators are equivalent in some sense. References [21–23] give definitions for the concept of dynamic systems equivalency; in this work, the equivalence between dynamic systems is addressed based on the following definition, adapted from [23].

**Definition 1** *It is said that the systems*

$$\Sigma \quad : \quad \dot{\chi} = f(\chi) + g(\chi)v \tag{36}$$

$$\Pi \quad : \quad \dot{\widetilde{\chi}} = \widetilde{f}(\widetilde{\chi}) + \widetilde{g}(\widetilde{\chi})\widetilde{v}, \tag{37}$$

*with $\chi, \widetilde{\chi} \in \mathbb{R}^{\kappa_1}$, $v, \widetilde{v} \in \mathbb{R}^{\kappa_2}$ are equivalent if there exist:*

*i. A diffeomorphism*

$$\widetilde{\chi} = \Phi(\chi) \tag{38}$$

*ii. A static state feedback*

$$v = \alpha_u(\chi) + \beta_u(\chi)\widetilde{v}, \tag{39}$$

*with $\beta_u(\chi)$ a nonsingular square matrix, such that the transformation of $\Sigma$ under $(\Phi, \alpha_u, \beta_u)$ is equal to $\Pi$.*

Fig 7 shows the link variables $\gamma_1$ and $\gamma_2$ of the R1, R2 manipulator. Hence, for the dynamic systems addressed in this work, Definition 1 can be applied considering that the system $\Sigma$

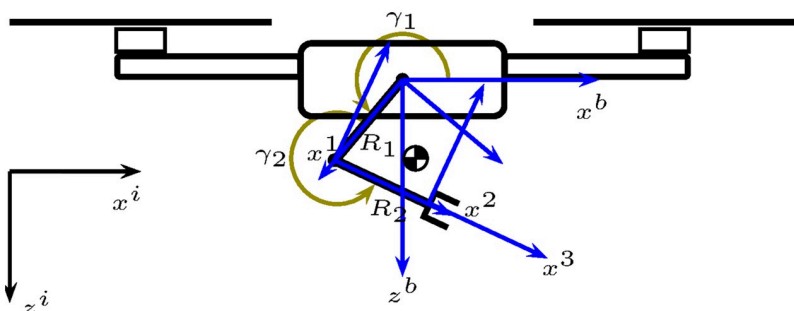

**Fig 7. Angles $\gamma_1$ and $\gamma_2$ on green.**

corresponds to the two revolute joints manipulator; thus, $\chi = [\gamma_1 \; \gamma_2 \; \dot{\gamma}_1 \; \dot{\gamma}_2]^\top$, $v = [\tau_1 \; \tau_2]^\top$. At the same time, $\Pi$ is the revolute-prismatic joints manipulator system; this is $\widetilde{\chi} = [\theta_P \; L \; \dot{\theta}_P \; \dot{L}]^\top$, $\bar{v} = [\bar{\tau}_R \bar{f}_P]^\top$.

It is possible to verify that the diffeomorphism (38) and the static state feedback (39) can be defined as follows

$$
\begin{bmatrix} \theta_P \\ L \\ \dot{\theta}_P \\ \dot{L} \end{bmatrix} = \Phi(\gamma_1, \gamma_2, \dot{\gamma}_1, \dot{\gamma}_2) \tag{40}
$$

$$
\begin{bmatrix} \tau_1 \\ \tau_2 \end{bmatrix} = \beta_u(\gamma_1, \gamma_2) \begin{bmatrix} \bar{\tau}_R \\ \bar{f}_P \end{bmatrix}
$$

where

$$
\Phi(\gamma_1, \gamma_2, \dot{\gamma}_1, \dot{\gamma}_2) = \begin{bmatrix} \frac{1}{2}\sqrt{d_l} \\[2mm] \frac{\pi}{2} + \gamma_1 + \arcsin\left(\frac{l_2 \sin(\gamma_2)}{\sqrt{d_l}}\right) \\[4mm] -\frac{l_1 l_2 s_{\gamma_2} \dot{\gamma}_2}{\sqrt{d_l}} \\[4mm] \dot{\gamma}_1 - \frac{\dot{\gamma}_2 l_2 (2c_{\gamma_2}^2 l_1 l_2 - d_l c_{\gamma_2} - 2l_1 l_2)}{d_l \sqrt{c_{\gamma_2}^2 l_2^2 - l_2^2 + d_l}} \end{bmatrix} \tag{41}
$$

$$
\beta_u(\gamma_1, \gamma_2) = \begin{bmatrix} 1 & 0 \\[2mm] \frac{l_2}{2L} \sin\left(\frac{\pi}{2} + \gamma_2 - \alpha\right) & \frac{l_2}{2} \sin(\alpha - \gamma_2) \end{bmatrix} \tag{42}
$$

with

$$
\alpha = \arcsin\left(\frac{l_2 \sin(\gamma_2)}{\sqrt{d_l}}\right) \text{ and } d_l = 4l_1^2 + l_2^2 + 4l_1 l_2 c_{\gamma_2}.
$$

**Table 2. UAM physical parameters.**

| Parameter | Value |
|---|---|
| $M$ | $1\ Kg$ |
| $m_1$ | $0.1\ Kg$ |
| $m_2$ | $0.1\ Kg$ |
| $l_1$ | $0.4\ m$ |
| $l_2$ | $0.4\ m$ |
| Links arm inertia | $0.0001\ Kg \cdot m^2$ |
| UAV inertia | $0.048\ Kg \cdot m^2$ |

The simulator considers a scenario where the UAM must fly close to target position in the proximity of a virtual object, after 10 seconds the end effector follows a desired trayectory to achive a desired final position. Thus, given a desired end-effector reference position, $x_{EEd}$, $z_{EEd}$, the desired angular positions $\gamma_{1d}$ and $\gamma_{2d}$ are obtained from the inverse kinematics and using the diffeomorphism (40), one gets

$$\begin{bmatrix} \theta_{Pd} \\ L_d \\ \dot{\theta}_{Pd} \\ \dot{L}_d \end{bmatrix} = \Phi(\gamma_{1d}, \gamma_{2d}, \dot{\gamma}_{1d}, \dot{\gamma}_{2d}),$$

The physical paramethers are sumarized on Table 2, while the gains values are on Table 3.

The diagram in Fig 8 illustrates how the simulations were implemented for the manipulator.

The realistic simulator where the simulations were implemented enables disturbances over the system generated through a wind profile. The wind profile can be user-defined or generated from random data. Thus, to evaluate the performance of the proposed control strategy, five simulations were driven, each with a different random wind disturbance profile, as seen in Fig 9. All Figures containing data plots were generated with the tool Professional Plots [24]. Table 4 shows the wind magnitude and direction media values for each simulation.

**Table 3. UAM controller gains.**

| Gain | Value |
|---|---|
| $\Gamma_1$ | 5.5 |
| $\Gamma_2$ | 5.5 |
| $\Gamma_3$ | 10.5 |
| $\Gamma_4$ | 60 |
| $K_{PX}$ | diag{4.5, 6.2} |
| $K_{DX}$ | diag{5, 6.2} |
| $K_{PQ}$ | 125 |
| $K_{DQ}$ | 90 |
| $K_{PA}$ | 25 |
| $K_{PP}$ | 50 |
| $K_{DA}$ | 25 |
| $K_{DP}$ | 50 |

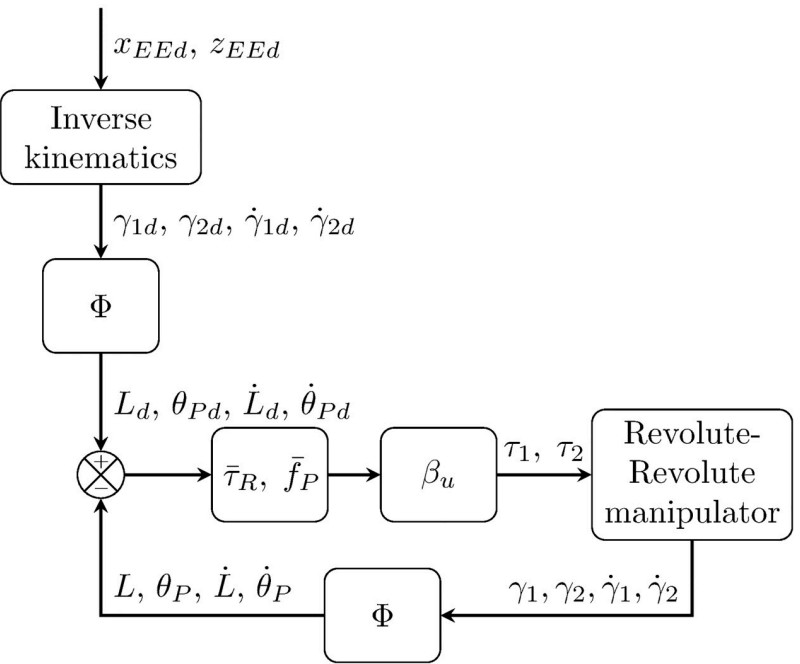

**Fig 8. Robot arm simulation implementation.**

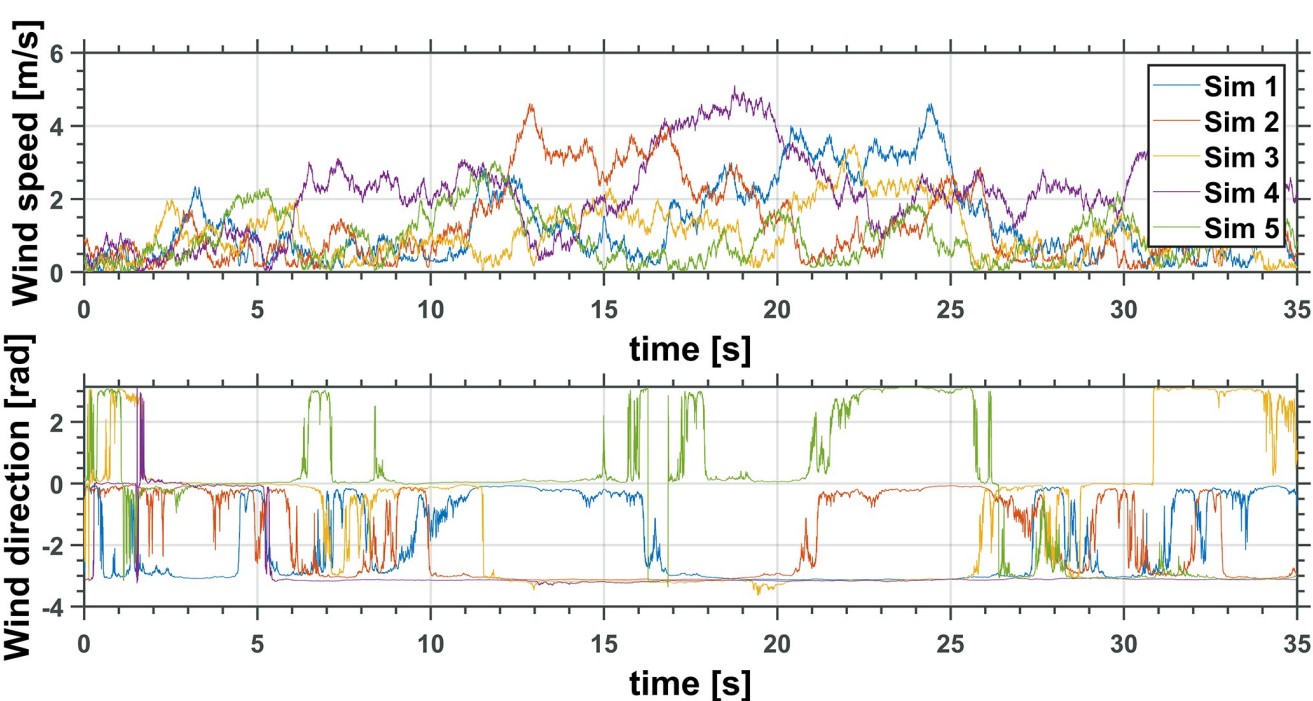

**Fig 9. Wind disturbance profile defined by wind magnitude and direction.**

**Table 4. Wind profile media magnitude and direction for simulations 1 to 5.**

| Simulation | Wind direction media [rad] | Wind magnitude media [m/s] |
|---|---|---|
| 1 | -1.9294 | 1.3020 |
| 2 | -1.7974 | 1.3070 |
| 3 | -1.0304 | 1.0674 |
| 4 | -2.6946 | 2.2006 |
| 5 | -0.1799 | 0.9544 |

Figs 10 and 11 depict the time evolution of the UAV translational axis errors, $\widetilde{x}$ and $\widetilde{z}$, respectively, while Fig 12 depicts the UAV attitude error, $\widetilde{\theta}_Q$.

Figs 13 and 14 show the end effector position error $\widetilde{x}_{EE} = x_{EE} - x_{EEd}$ and $\widetilde{z}_{EE} = z_{EE} - z_{EEd}$, respectively. As can be observed, all error signals converge to a zero neighborhood, as the theoretical analysis predicted.

The equivalent robotic manipulator errors $\widetilde{L}$ and $\widetilde{\theta}_P$ are shown in Figs 15 and 16, while the original robotic manipulator errors $\widetilde{\gamma}_1 = \gamma_1 - \gamma_{1d}$ and $\widetilde{\gamma}_2 = \gamma_2 - \gamma_{2d}$ are reported in Figs 17 and 18. Note that the errors on the equivalent robotic manipulator are closer to zero than those from the original. This behavior can be caused by unknown parameters implemented on the realistic simulator.

Figs 19 and 20 show the control inputs on the UAM only for the first simulation. The following integral functions were measured for each simulation to understand the controller

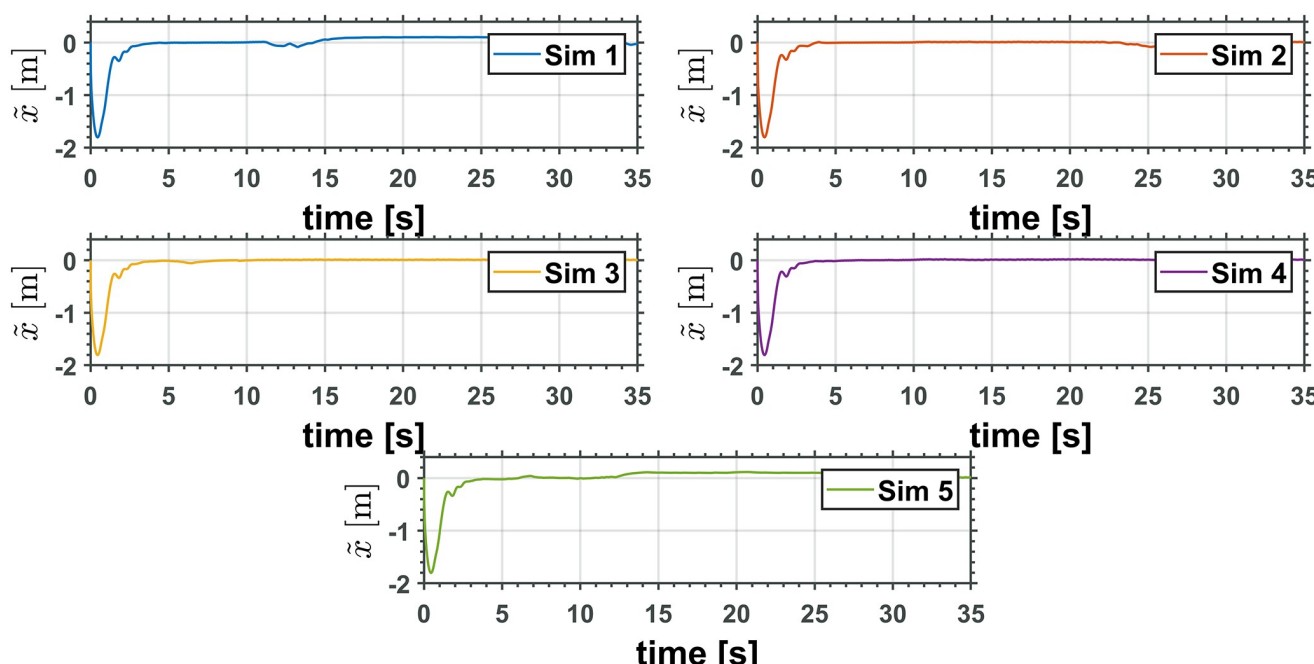

**Fig 10. UAV translational *x* axis error.**

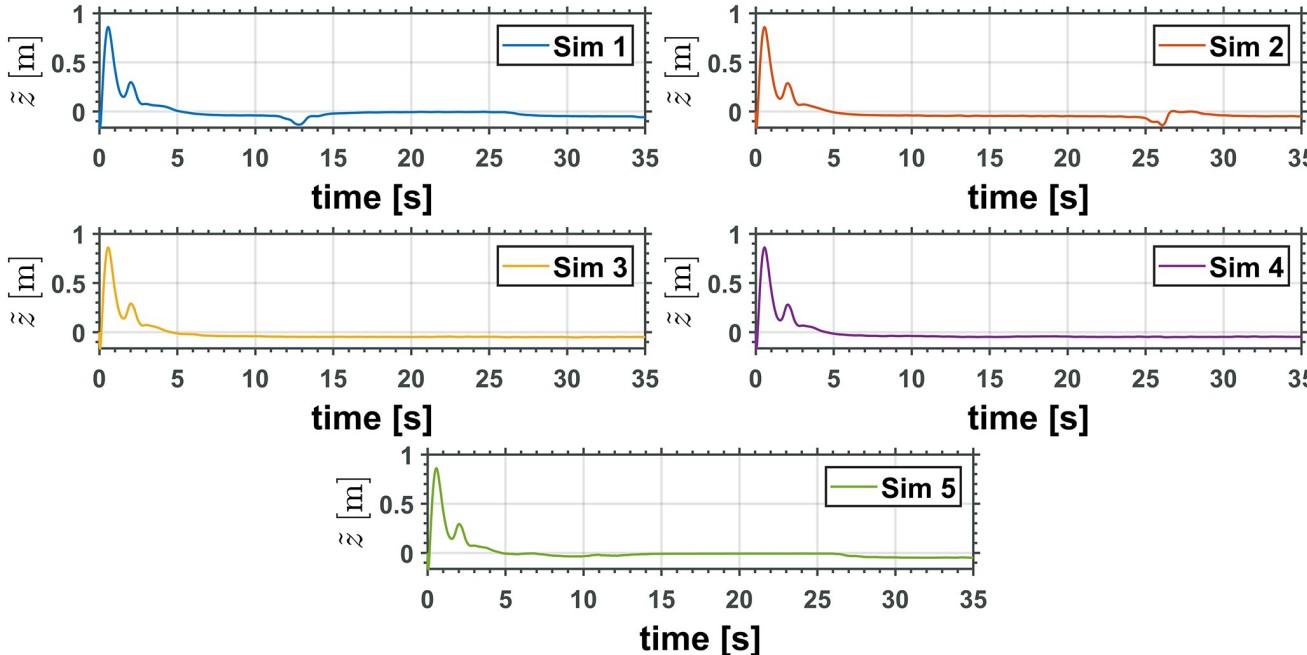

**Fig 11. UAV translational $z$ axis error.**

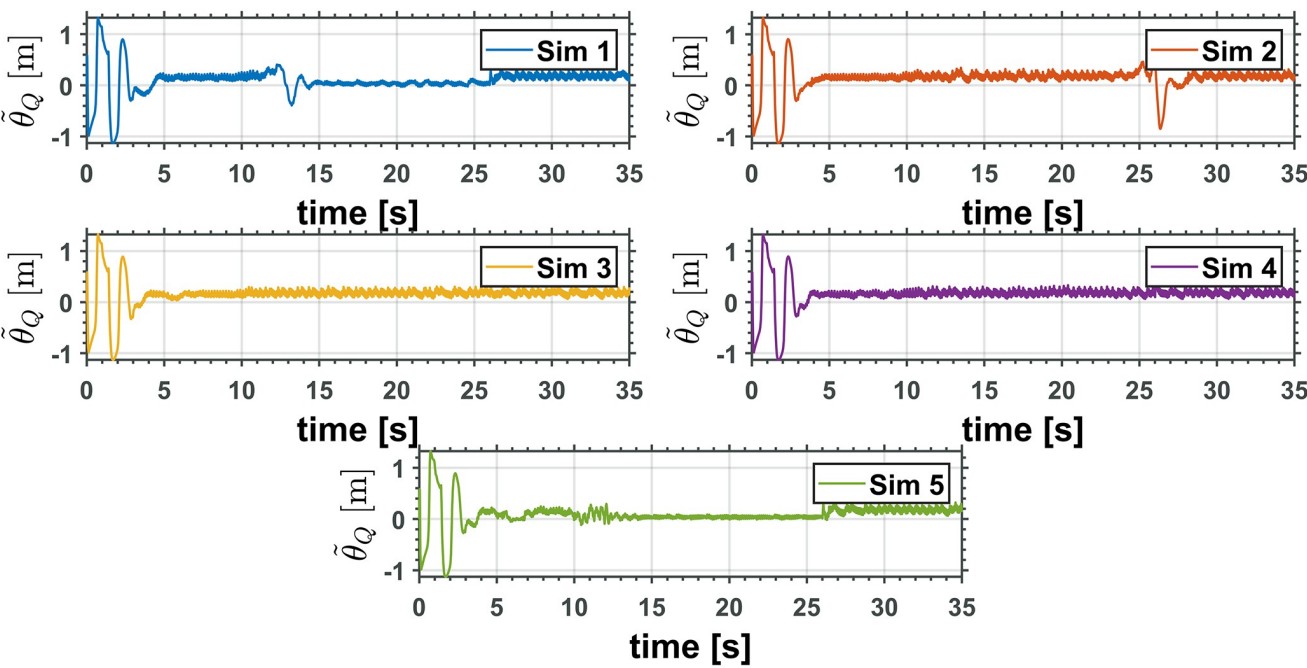

**Fig 12. UAV attitude error.**

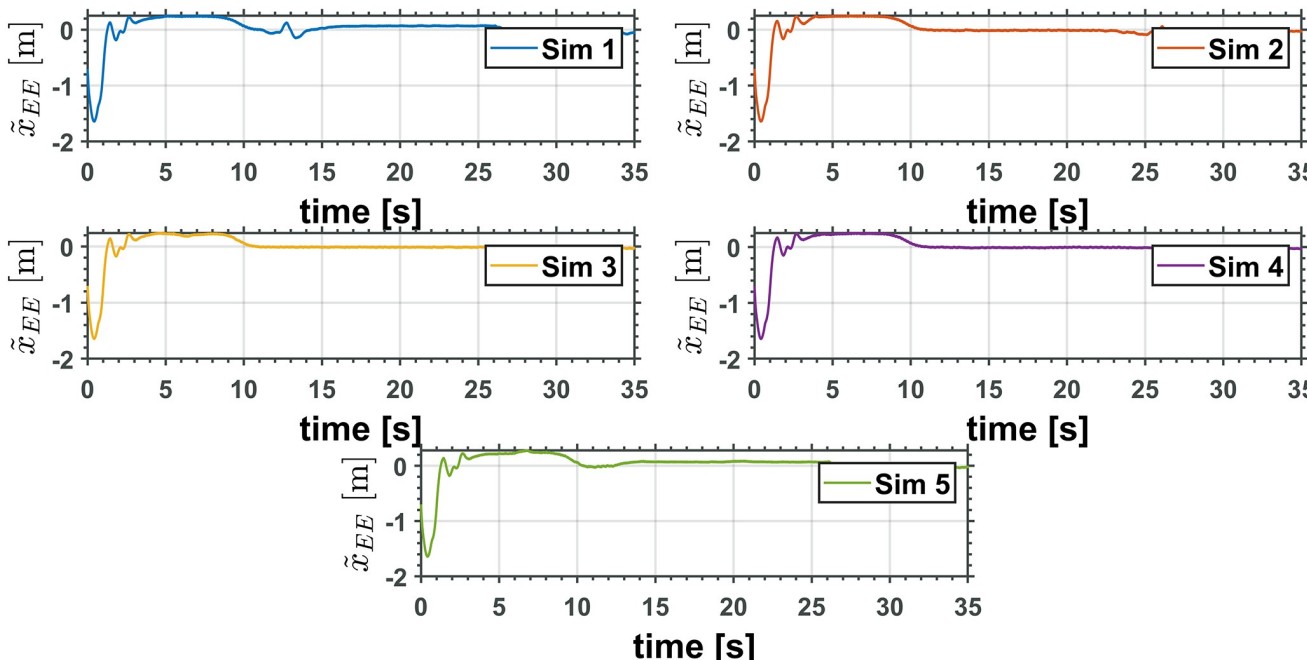

**Fig 13. End effector *x* axis position error.**

performance in all simulations better,

$$
\begin{aligned}
F_1 &= \int_0^t \| \widetilde{X}(\tau) \| \, d\tau \\
F_2 &= \int_0^t \widetilde{\theta}_Q(\tau)^2 d\tau \\
F_3 &= \int_0^t \left( \widetilde{\gamma}_1(\tau)^2 + \widetilde{\gamma}_2(\tau)^2 \right) d\tau \\
F_4 &= \int_0^t \left( \widetilde{x}_{EE}(\tau)^2 + \widetilde{z}_{EE}(\tau)^2 \right) d\tau \\
F_5 &= \int_0^t \left( \widetilde{L}(\tau)^2 + \widetilde{\theta}_P(\tau)^2 \right) d\tau
\end{aligned}
\tag{43}
$$

Table 5 presents the values for each measurement $F_i$, $i = 1, \cdots, 5$ correspondent to each simulation.

From the values in Table 5, it can be concluded that the control performance remains the same for different wind profiles acting on the system as disturbances. Hence, the proposed disturbance estimator performs adequately. Fig 21 presents the disturbance estimated by the proposed estimation strategy for simulation 1.

Fig 22 shows the UAM sequence followed during the simulation. From number 1 to number 4, the UAM approaches a reference near the blue dot, the reference for the robotic manipulator, and remains in such a position. In number 5, the UAM is already on its reference so that the robotic arm can also reach its reference.

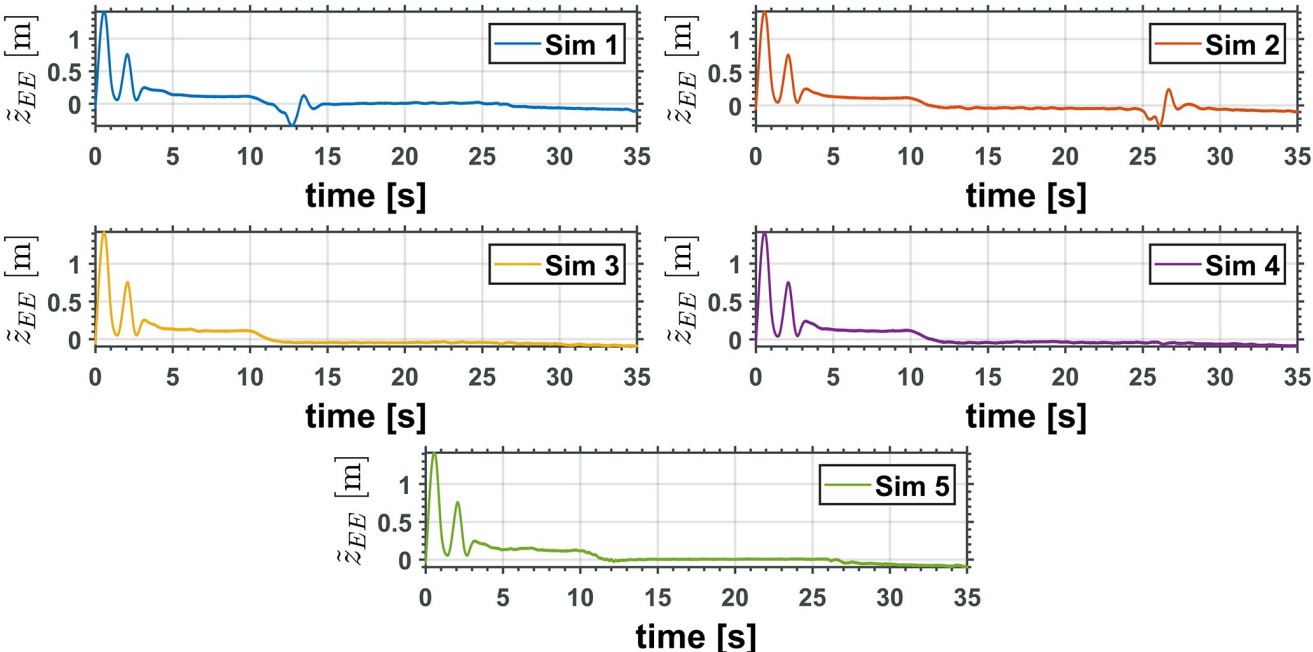

**Fig 14. End effector $z$ axis position error.**

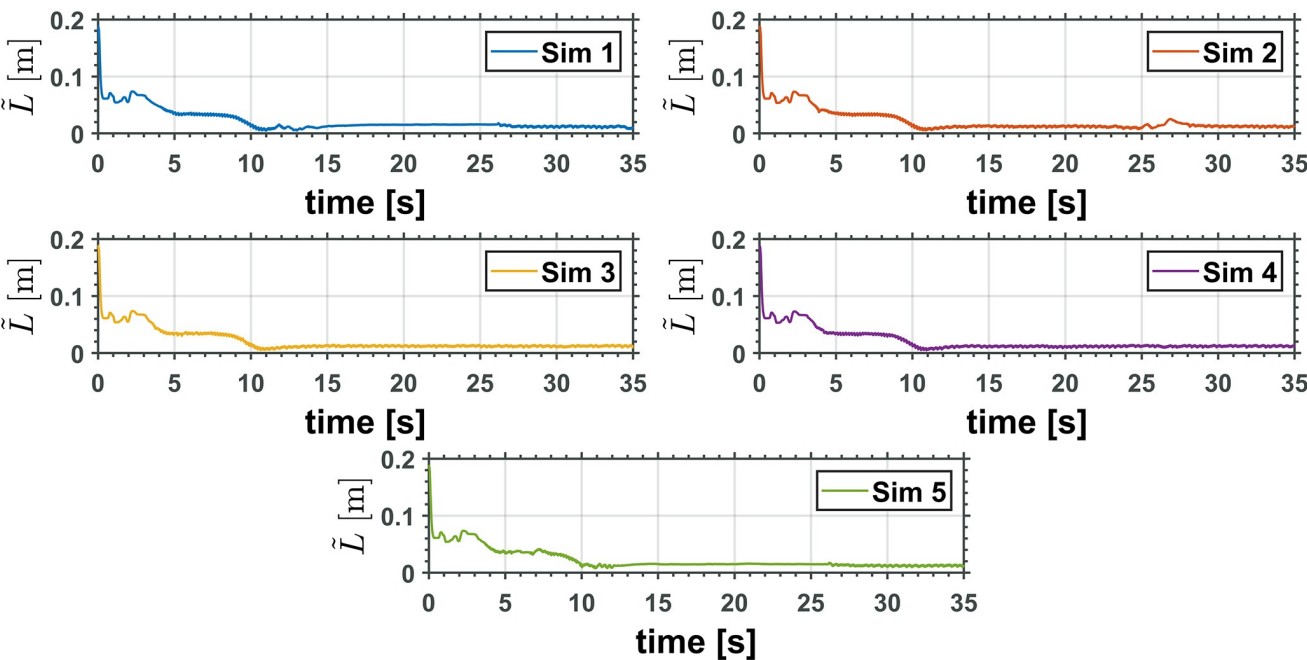

**Fig 15. Equivalent manipulator prismatic joint error, $\widetilde{L}$.**

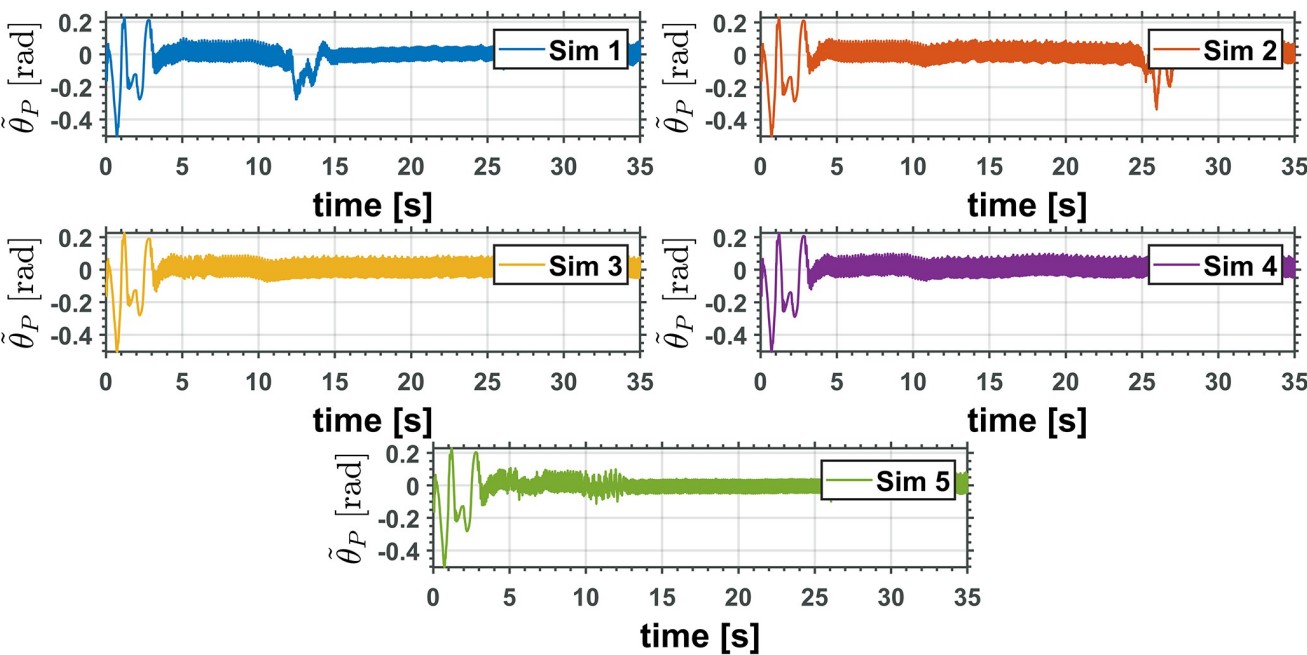

**Fig 16. Equivalent manipulator revolute joint error, $\widetilde{\theta}_P$.**

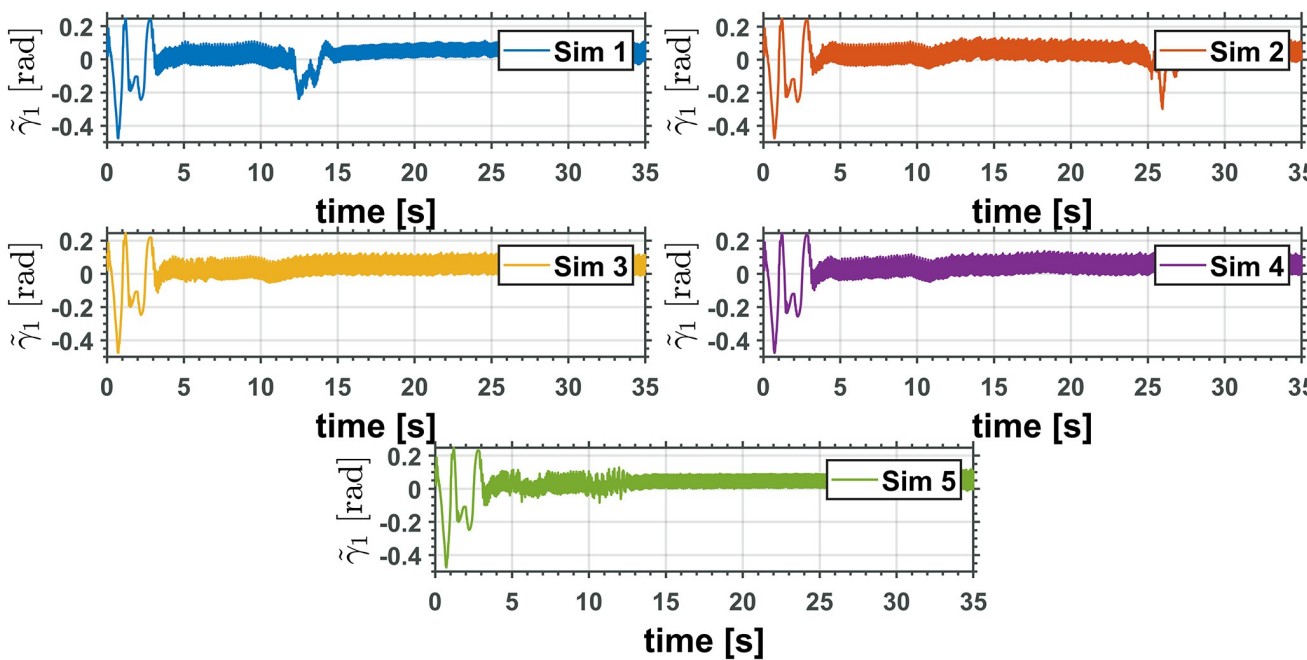

**Fig 17. Two revolute joints manipulator joint error, $\widetilde{\gamma}_1$.**

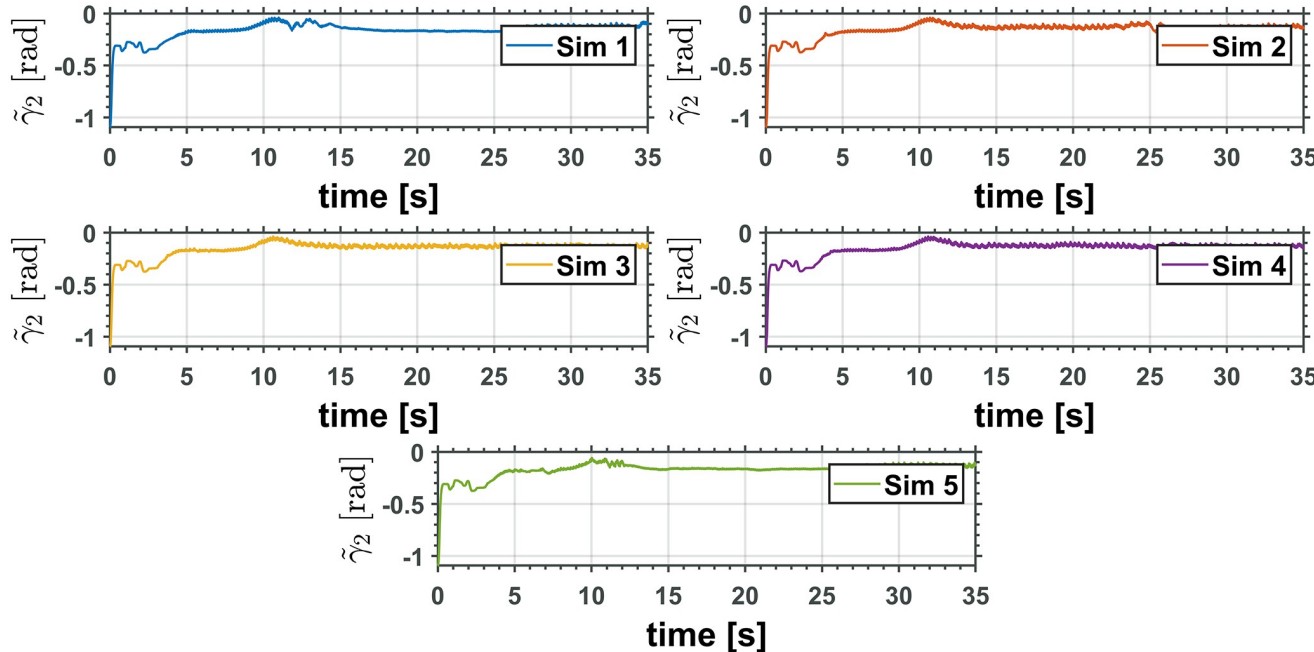

**Fig 18. Two revolute joints manipulator joint error, $\widetilde{\gamma}_2$.**

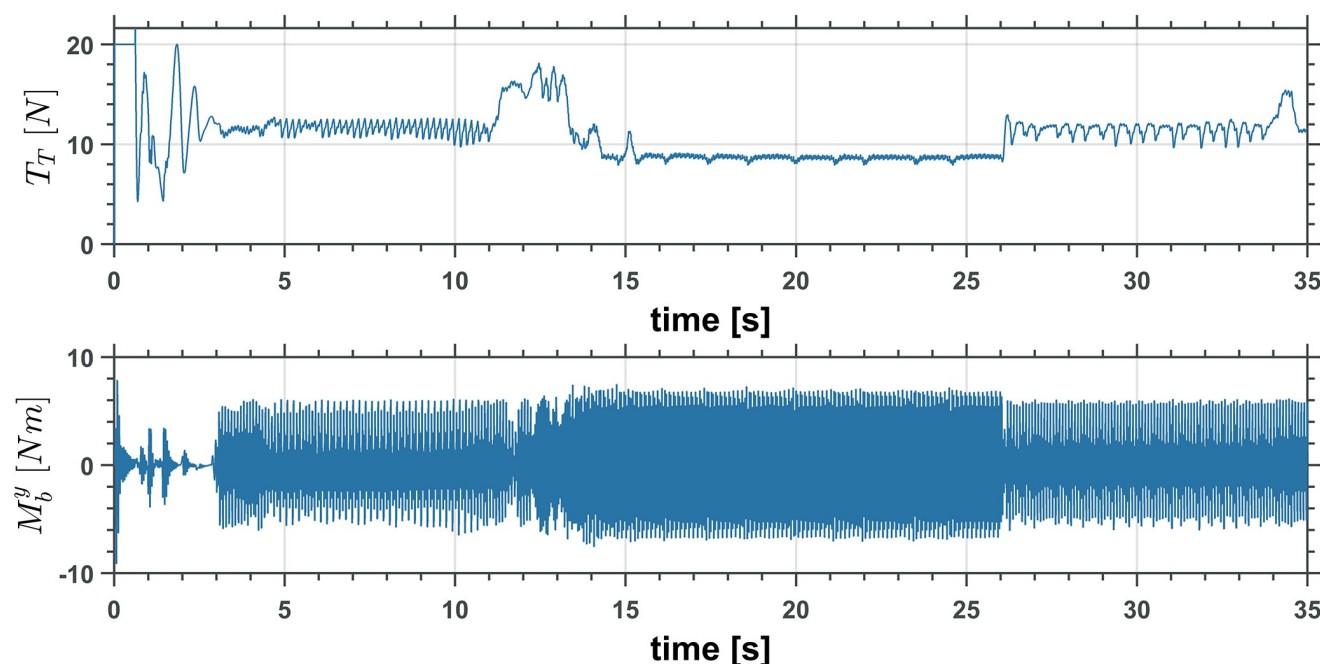

**Fig 19. UAV total trust, $T_T$ and moment, $\bar{M}_b^y$ input controls.**

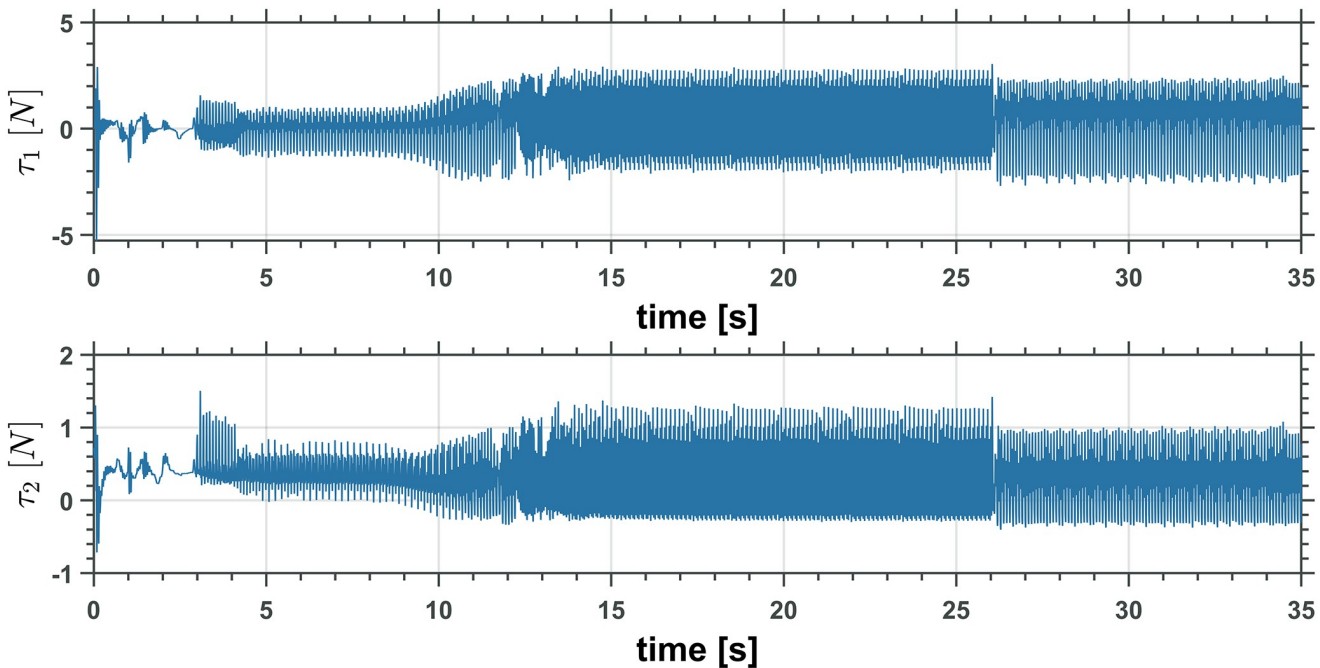

**Fig 20. Two revolute joints manipulator control inputs, $\tau_1$ and $\tau_2$.**

## 5 Conclusions

This work proposed a control algorithm for an Unmanned Aerial Manipulator. Analyzing the effects on a flying platform generated by a two-revolute manipulator was simplified using an equivalent revolute-prismatic joints manipulator. This approach permitted compensating for the known dynamics and decoupling the UAV dynamics from the remaining robotic manipulator dynamics. Thus, the remaining manipulator dynamics were treated as external forces and moments acting on the quadrotor. The resulting dynamical UAM structure permits designing a disturbance estimator based on the Immersion and Invariance technique. Then, a PD-like controller with disturbance compensation is proposed to solve the trajectory tracking problem for the UAM. A formal stability analysis of the resulting closed-loop dynamics is presented.

Numerical simulations in a realistic simulator are presented to evaluate the proposed control strategy. The realistic simulator considers wind profiles acting on the UAM. For future work, this work has established a solid base for an extension of the results to find the

**Table 5. Simulations with random wind disturbances, measuring function $F_i$, $i = 1, 2, 3, 4, 5$.**

| Simulation | $F_1$ | $F_2$ | $F_3$ | $F_4$ | $F_5$ |
|---|---|---|---|---|---|
| 1 | 4.42 | 6.19 | 6.29 | 6.03 | 2.40 |
| 2 | 3.78 | 7.96 | 6.11 | 5.76 | 2.48 |
| 3 | 3.76 | 7.73 | 5.90 | 5.40 | 2.27 |
| 4 | 3.67 | 7.85 | 5.87 | 5.37 | 2.29 |
| 5 | 4.35 | 5.33 | 6.48 | 5.77 | 2.21 |
| Average | 4.0011 | 7.0153 | 6.1364 | 5.6683 | 2.3364 |
| Standard deviation | 0.3597 | 1.1819 | 0.2591 | 0.2794 | 0.1091 |

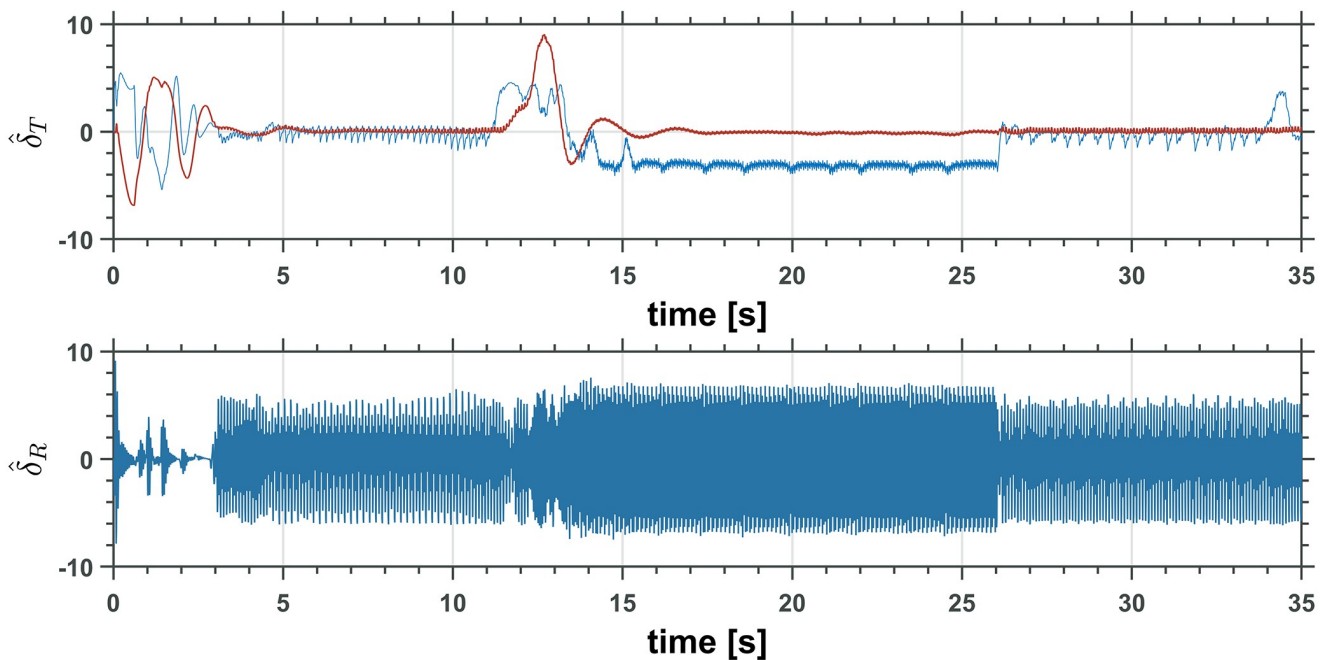

**Fig 21. Disturbances estimated by the proposed estimation strategy.**

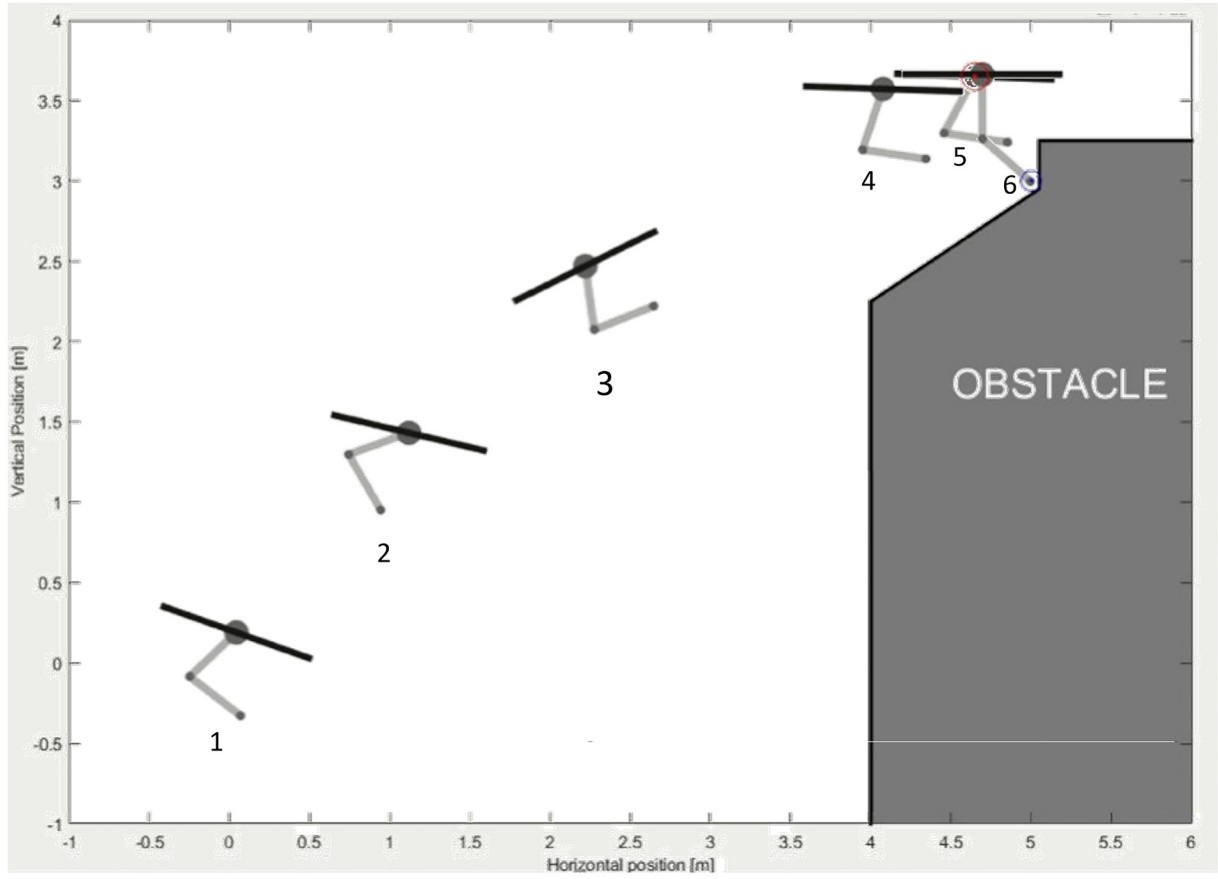

**Fig 22. Simulation sequence.**

equivalence between n-degrees of freedom revolute joints manipulator and an R-P type manipulator, this approach would simplify the disturbances analysis on more general UAM configurations.

## 6 Appendices

### A Recursive Newton-Euler Algorithm

In this work the RNEA reported in [12] is followed. The angular velocity propagation is computed from the following equations. For the revolute joint

$$\Omega^{i+1} = \binom{i+1}{i}R)\Omega^i + \dot{\theta}_{i+1}\hat{z}^{i+1} \tag{44}$$

and for the prismatic joint

$$\Omega^{i+1} = \binom{i+1}{i}R)\Omega^i \tag{45}$$

The angular and translational accelerations are propagated as follows. For a revolute joint

$$\dot{\Omega}^{i+1} = \binom{i+1}{i}R)\Omega^i + \binom{i+1}{i}R)\Omega^i \times \dot{\theta}_{i+1}\hat{z}^{i+1} + \ddot{\theta}_{i+1}\hat{z}^{i+1} \tag{46}$$

and

$$\dot{V}^{i+1} = \binom{i+1}{i}R)\left[\dot{\Omega}^i \times P^i_{i+1} + \Omega^i \times (\Omega^i \times P^i_{i+1}) + \dot{V}^i\right] \tag{47}$$

For a prismatic joint

$$\dot{\Omega}^{i+1} = \binom{i+1}{i}R)\Omega^i \tag{48}$$

and

$$\begin{aligned}
\dot{V}^{i+1} &= \binom{i+1}{i}R)\left[\dot{\Omega}^i \times P^i_{i+1} + \Omega^i \times (\Omega^i \times P^i_{i+1}) + \dot{V}^i\right] \\
&\quad + 2\Omega^{i+1} \times \dot{d}_{i+1}\hat{z}^{i+1} + \ddot{d}_{i+1}\hat{z}^{i+1}
\end{aligned} \tag{49}$$

Finally, the link center of mass acceleration is computed as

$$\dot{V}^{i+1}_C = \dot{\Omega}^{i+1} \times P^{i+1}_C + \Omega^{i+1} \times (\Omega^{i+1} \times P^{i+1}_C) + \dot{V}^{i+1} \tag{50}$$

and the forces and moments acting at the center of gravity are

$$\begin{aligned}
F^{i+1} &= m_{i+1}\dot{V}^{i+1}_C \\
N^{i+1} &= J_{i+1}\dot{\Omega}^{i+1} + \Omega^{i+1} \times J_{i+1}\Omega^{i+1}
\end{aligned} \tag{51}$$

This completes the outward RNEA iteration.

The inward RNEA starts computing the forces acting on each link as

$$\begin{aligned}
f^i &= F^i + \binom{i}{i+1}R)f^{i+1} \\
n^i &= N^i + \binom{i}{i+1}R)n^{i+1} + P^i_C \times F^i + P^i_{i+1} \times \binom{i}{i+1}R)f^{i+1}
\end{aligned} \tag{52}$$

## Author Contributions

**Conceptualization:** Yarai Elizabeth Tlatelpa-Osorio.

**Data curation:** Yarai Elizabeth Tlatelpa-Osorio.

**Formal analysis:** Yarai Elizabeth Tlatelpa-Osorio, Hugo Rodríguez-Cortés, J. Á. Acosta.

**Investigation:** Yarai Elizabeth Tlatelpa-Osorio, Hugo Rodríguez-Cortés, J. Á. Acosta.

**Methodology:** Hugo Rodríguez-Cortés, J. Á. Acosta.

**Project administration:** Hugo Rodríguez-Cortés, J. Á. Acosta.

**Software:** J. Á. Acosta.

**Supervision:** Hugo Rodríguez-Cortés, J. Á. Acosta.

**Validation:** Yarai Elizabeth Tlatelpa-Osorio.

**Visualization:** Yarai Elizabeth Tlatelpa-Osorio.

**Writing – original draft:** Yarai Elizabeth Tlatelpa-Osorio, Hugo Rodríguez-Cortés, J. Á. Acosta.

**Writing – review & editing:** Hugo Rodríguez-Cortés, J. Á. Acosta.

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
