## [Decision Letter · Decision Letter 0]

28 Nov 2023

PONE-D-23-34692A decentralized approach for the aerial manipulator robust trajectory tracking.PLOS ONE

Dear Dr. Tlatelpa-Osorio,

Thank you for submitting your manuscript to PLOS ONE. After careful consideration, we feel that it has merit but does not fully meet PLOS ONE’s publication criteria as it currently stands. Therefore, we invite you to submit a revised version of the manuscript that addresses the points raised during the review process. Please submit your revised manuscript by Jan 12 2024 11:59PM. If you will need more time than this to complete your revisions, please reply to this message or contact the journal office at plosone@plos.org. Please include the following items when submitting your revised manuscript:A rebuttal letter that responds to each point raised by the academic editor and reviewer(s). You should upload this letter as a separate file labeled 'Response to Reviewers'.A marked-up copy of your manuscript that highlights changes made to the original version. You should upload this as a separate file labeled 'Revised Manuscript with Track Changes'.An unmarked version of your revised paper without tracked changes. You should upload this as a separate file labeled 'Manuscript'.If applicable, we recommend that you deposit your laboratory protocols in protocols.io to enhance the reproducibility of your results. Protocols.io assigns your protocol its own identifier (DOI) so that it can be cited independently in the future. For instructions see: https://journals.plos.org/plosone/s/submission-guidelines#loc-laboratory-protocols. Additionally, PLOS ONE offers an option for publishing peer-reviewed Lab Protocol articles, which describe protocols hosted on protocols.io. Read more information on sharing protocols at https://plos.org/protocols?utm_medium=editorial-email&utm_source=authorletters&utm_campaign=protocols.

We look forward to receiving your revised manuscript.

Kind regards,

Gang Wang

Academic Editor

PLOS ONE

Journal Requirements:

"Y.E. Tlatelpa-Osorio

702178

CONSEJO NACIONAL DE CIENCIA Y TECNOLOGÍA

 " ext-link-type="uri" xlink:type="simple">https://conahcyt.mx/" 

Additional Editor Comments:

Some minor revisions should be made based on the reviewers' suggestions before acceptance.

Reviewers' comments:

Reviewer's Responses to Questions

**Comments to the Author**

1. Is the manuscript technically sound, and do the data support the conclusions?

Reviewer #1: Yes

Reviewer #2: Yes

2. Has the statistical analysis been performed appropriately and rigorously? 

Reviewer #1: Yes

Reviewer #2: Yes

3. Have the authors made all data underlying the findings in their manuscript fully available?

Reviewer #1: Yes

Reviewer #2: No

4. Is the manuscript presented in an intelligible fashion and written in standard English?

Reviewer #1: Yes

Reviewer #2: Yes

5. Review Comments to the Author

Reviewer #1: This paper aims to propose a new control strategy for an unmanned aerial manipulator (UAM). First, an equivalent model is proposed to equate the 2-revolute links robotic manipulator model to the 1-revolute, 1-prismatic links robotic manipulator model. Then based on this equivalent model, this paper designs a distributed controller for UAW for tracking. After careful reading, I have the following concerns.

1. The background introduction of the article is very detailed, but it seems that it is not clearly stated in the contribution section of the article which problems existing in the above-mentioned existing work are solved by this paper.

2. The structure of the article is quite messy, and I didn't find any basis for such an equivalent transformation. I also found no description of how this equivalence method simplifies the stability analysis.

3. In the section describing the UAM dynamics model, it is written that this paper considers a robotic manipulator to have two revolute joints, but in the simulation section, it is stated that the proposed controller is designed considering a quadrotor with a manipulator composed of a revolute joint and a prismatic joint.

4. The description of the simulation scene settings in the simulation part of the paper is insufficient, and the figures of the simulation scenes do not show what the author wants to prove.

5. The writing format of this paper is very irregular, especially the references section.

Reviewer #2: This article proposes a Decentralized Robust Control Strategy for unmanned aerial manipulators in the presence of external disturbances. The author proposes an equivalent dynamic model that facilitates analysis and can be used to analyze the complete properties of the system. Overall, this article has sufficient workload and also provides a complete analysis and simulation process. Here are my suggestions.

1.The paper fail to mention the differences and advantages and disadvantages between centralized and decentralized control, and why decentralized control strategies should be adopted.

2.The format of the paper still needs to be checked and modified. For example, in the last paragraph of the Introduction, the Roman numerals after the section cannot be displayed.

3.The images in the paper are very blurry, and I cannot see the parameters of UAV and other values in the last image clearly.

6. PLOS authors have the option to publish the peer review history of their article (what does this mean?). If published, this will include your full peer review and any attached files.

Reviewer #1: No

Reviewer #2: No

---

## [Author Response · Author response to Decision Letter 0]

17 Jan 2024

*Please see the document "Response to Reviewers", there the format is clearer. Thank you in advance.

Reply to Rewiever #1

 R1: This paper aims to propose a new control strategy for an unmanned aerial

manipulator (UAM). First, an equivalent model is proposed to equate the 2-revolute links robotic

manipulator model to the 1-revolute, 1-prismatic links robotic manipulator model. Then based on

this equivalent model, this paper designs a distributed controller for UAM for tracking. }

 Comment. Thanks for the time you devoted to revise our work. 

R1: After careful

reading, I have the following concerns.

1. The background introduction of the article is very detailed, but it seems that it is not clearly stated in the contribution section of the article which problems existing in the above-mentioned existing work are solved by this paper.} 

Response. Thanks for this comment, which made us realize we

needed to clarify this contribution in the manuscript. We have rewritten part of the introduction section to highlight the problem with previous works and our contribution. 

We notice that in most works addressing the Aerial Manipulator control by the decentralized approach, the interaction between the two dynamic sub-systems, robot arm, and UAV, is not studied, just neglected or somehow compensated, either it is only assumed that the dynamic interaction between the manipulator and the aerial vehicle can be treated as exogenous disturbances acting on each sub-system or neglected. Thus, as far as we know, the only work theoretically treating the interaction is in previos works, but only at the kinematic level of the robot arm, which means that, in practice, the analysis is only valid for slow movements when the accelerations can be neglected.

This interaction is essential as the UAV dynamics are a part of the robotic arm dynamics, and the Newton-Euler Recursive algorithm helped us clarify it. Thus, in our proposed contribution, we deeply analyze the interaction at a dynamic level, not just at the robot arm kinematics, as in \\cite{acosta2020accurate}. This study has two main new outcomes: 1) being able to demonstrate that the interaction between the two subsystems can be treated as exogenous perturbations by proposing a novel equivalent transformation that allows designing an estimator for the torques and forces of the interaction, and 2) the estimator paves the way to design a controller capable of tracking prescribed trajectories with exponential convergence, a robust property highly desirable in practice. We emphasize that this analysis is missing in the literature, normally avoided with assumptions such as slow motion, which are not practical because there may be external disturbances such as gusts of wind that cause high accelerations. Indeed, in the simulation section, we show good performance under unexpected external wind gusts, thanks to the in-the-loop estimator.

2. The structure of the article is quite messy, and I didn't find any basis for such an equivalent transformation. I also found no description of how this equivalence method simplifies the stability analysis.

Response. Thanks for your comments. The content is reorganized, and some of the section’s names have been modified as follows.

Section: UAM dynamic model.

 Subsection: Quadrotor dynamic model.

 Subsection: Robotic manipulator dynamic model.

Section: Decentralized robust control strategy.

 Subsection: Quadrotor exogeneous disturbances estimator. 

 Subsection: Quadrotor position and attitude control.

 Subsection:Robot arm controller based on the equivalent model.

 Subsection: UAM closed loop dynamics. Subsection: Stability analysis.

Section: Numerical simulations.

In the beginning of Section Decentralized robust control strategy, we have added the following paragraph:

The control design is divided into two control loops: the inner loop is a state feedback controller that customizes the robotic manipulator effects on the aerial vehicle as an exogenous disturbance (C4D), generating a decentralized UAM dynamic model. In contrast, the outer control loop uses the decentralized model and independently applies control strategies for the quadrotor and robotic arm. We first present the C4D state feedback and then the estimator of the quadrotor exogeneous moments and forces (EQEMF). 

We have re-state a phrase to better clarify the difference between the two closed-loop systems, from:

The UAM closed-loop dynamics (10)–(12) with the controller (13)–(15) results into:

' The UAM dynamics (10)–(12) with the inner loop controller RC4D (13)–(15) results in ... '

In regards of the comment about the stability analysis simplification we can say that the proposed coordinate transformation reveals the exogenous nature of the interaction forces in the equivalent dynamics, and therefore enables the design of the estimator and the subsequent stability analysis.

3. In the section describing the UAM dynamics model, it is written that this paper considers a robotic manipulator to have two revolute joints, but in the simulation section, it is stated that the proposed controller is designed considering a quadrotor with a manipulator composed of a revolute joint and a prismatic joint.

Response. We thank this reviewer for this comment; after re-reading it, we agree that the explanation is unclear. One of the main contributions of this work is that we re-formulate the two revolute joints manipulator (1R2R) by a prismatic and a revolute joints (1R1P). Then, we design the control strategy for the 1R1P manipulator. Therefore, using the diffeomorphism transformation of Definition 1 in the simulations section, we can ‘mathematically translate’ the designed control strategy on the 1R2R joint manipulator for its implementation. Although this step might seem straightforward, it has been included for clarity of presentation.

4. The description of the simulation scene settings in the simulation part of the paper is insufficient, and the figures of the simulation scenes do not show what the author wants to prove.

Response. Thanks for the comments, now we added more information about the simulation scenario as well as an improvement on the figures presented.

 5. The writing format of this paper is very irregular, especially the references section.

 Response. Thank you. We have carried out a complete revision of the manuscript to improve the reading. 

Reply to Reviewer #2

This article proposes a Decentralized Robust Control Strategy for unmanned aerial

manipulators in the presence of external disturbances. The author proposes an equivalent dynamic

model that facilitates analysis and can be used to analyze the complete properties of the system.

Overall, this article has sufficient workload and also provides a complete analysis and simulation

process.

Comment. Thank you for the time you devoted to reviewing our work and for highlighting the key points and the overall workload assessment.\\\\

Here are my suggestions.

1.The paper fail to mention the differences and advantages and disadvantages between centralized

and decentralized control, and why decentralized control strategies should be adopted.

 Response.

Thanks for your comment. In the introduction section, we have pointed out a reference that highlights the characteristics of the centralized and decentralized approaches. There is no crucial reason to follow one method or the other; however, the decentralized approach fits better with the attributes of the proposed disturbance estimator.

 2.The format of the paper still needs to be checked and modified. For example, in the last

paragraph of the Introduction, the Roman numerals after the section cannot be displayed.

 Response. Thanks for this observation, We carried out a complete revision of the manuscript and attended the problems with the format.

3.The images in the paper are very blurry, and I cannot see the parameters of UAV and other values

in the last image clearly.

Response. We changed the last image and added two tables, one with the physical parameters and other with the controller gains. Also, the quality of the images containing the error dynamics time evolution were revised.

---

## [Decision Letter · Decision Letter 1]

6 Feb 2024

A decentralized approach for the aerial manipulator robust trajectory tracking.

PONE-D-23-34692R1

Dear Dr. Tlatelpa-Osorio,

We’re pleased to inform you that your manuscript has been judged scientifically suitable for publication and will be formally accepted for publication once it meets all outstanding technical requirements.

Kind regards,

Gang Wang

Academic Editor

PLOS ONE

Additional Editor Comments (optional):

The reviewers' remaining concerns have been resolved and the revised paper is recommended for acceptance.

Reviewers' comments:

Reviewer's Responses to Questions

**Comments to the Author**

1. If the authors have adequately addressed your comments raised in a previous round of review and you feel that this manuscript is now acceptable for publication, you may indicate that here to bypass the “Comments to the Author” section, enter your conflict of interest statement in the “Confidential to Editor” section, and submit your "Accept" recommendation.

Reviewer #1: All comments have been addressed

2. Is the manuscript technically sound, and do the data support the conclusions?

Reviewer #1: Yes

3. Has the statistical analysis been performed appropriately and rigorously? 

Reviewer #1: Yes

4. Have the authors made all data underlying the findings in their manuscript fully available?

Reviewer #1: Yes

5. Is the manuscript presented in an intelligible fashion and written in standard English?

Reviewer #1: Yes

6. Review Comments to the Author

Reviewer #1: The authors carefully revised the manuscript in accordance with the reviewers' comments.

The paper describes the innovations clearly, the theoretical derivation is adequate and the simulation data are sufficient.

7. PLOS authors have the option to publish the peer review history of their article (what does this mean?). If published, this will include your full peer review and any attached files.

Reviewer #1: No

---

## [Editor Report · Acceptance letter]

26 Feb 2024

PONE-D-23-34692R1 

PLOS ONE

Dear Dr. Tlatelpa-Osorio, 

I'm pleased to inform you that your manuscript has been deemed suitable for publication in PLOS ONE. Congratulations! Your manuscript is now being handed over to our production team.

Kind regards, 

on behalf of

Dr. Gang Wang 

Academic Editor

PLOS ONE